# Stereotyped spatial patterns of functional synaptic connectivity in the cerebellar cortex

Antoine M Valera[1†], Francesca Binda[1], Sophie A Pawlowski[1‡], Jean-Luc Dupont[1], Jean-François Casella[1], Jeffrey D Rothstein[2], Bernard Poulain[1], Philippe Isope[1*]

[1]Institut des Neurosciences Cellulaires et Intégratives, CNRS Université de Strasbourg, Strasbourg, France; [2]Brain Science Institute, Johns Hopkins University, School of Medicine, Baltimore, United States

**Abstract** Motor coordination is supported by an array of highly organized heterogeneous modules in the cerebellum. How incoming sensorimotor information is channeled and communicated between these anatomical modules is still poorly understood. In this study, we used transgenic mice expressing GFP in specific subsets of Purkinje cells that allowed us to target a given set of cerebellar modules. Combining in vitro recordings and photostimulation, we identified stereotyped patterns of functional synaptic organization between the granule cell layer and its main targets, the Purkinje cells, Golgi cells and molecular layer interneurons. Each type of connection displayed position-specific patterns of granule cell synaptic inputs that do not strictly match with anatomical boundaries but connect distant cortical modules. Although these patterns can be adjusted by activity-dependent processes, they were found to be consistent and predictable between animals. Our results highlight the operational rules underlying communication between modules in the cerebellar cortex.

**\*For correspondence:** philippe.isope@inci-cnrs.unistra.fr

**Present address:** [†]Department of Neuroscience, Physiology and Pharmacology, University College London, London, United Kingdom; [‡]Department Pharmacology & Therapeutics, McGill University, Montréal, Canada

**Competing interests:** The authors declare that no competing interests exist.

## Introduction

Long-range connectivity between brain areas and the basic organization of microcircuits has been widely studied in many brain structures (*Shepherd and Grillner, 2010*). However, the synaptic connectivity at the mesoscale level, that is, the precise functional synaptic arrangement between cortical modules, has been described in few brain areas since module boundaries are often ill defined. In the rodent barrel cortex and visual cortex (*Callaway and Katz, 1993*; *Briggs and Callaway, 2001*; *Shepherd et al., 2003*), anatomical modules contain a wide array of neuronal types that interconnect within complex neuronal circuits (*Harris and Mrsic-Flogel, 2013*; *Cossell et al., 2015*) and reveal inter-modular communication that might influence the output of individual modules (e.g. see multi-whisker stimuli; *Estebanez et al., 2012*). However, whether neighboring modules strictly follow the same functional and cellular organization rules or display subtle differences in local architecture or neurochemical expression that determine module-specific function or processing rules is not established.

In the cerebellar cortex, where anatomical modules are well defined and are associated with different features of motor control, the crystalline structure of the cortex initially led to the concept that different functions arose through distinct patterns of input/output connectivity as observed in the barrel cortex with a given vibrissa targeting one specific barrel (*Ito, 1984*). However, much anatomical and functional evidence suggests that cerebellar modules consist of an array of microcircuits, each of which has a specific identity (*Scott, 1963*; *Lange et al., 1982*; *Brochu et al., 1990*; *Dehnes et al., 1998*; *Mateos et al., 2000*; *Wadiche and Jahr, 2005*; *Sarna et al., 2006*; *Shin et al.,*

**eLife digest** The human brain is essentially organised into modules made from groups of connected neurons. A part of the brain at the back of the head, called the cerebellum, is organised into particularly well-defined modules and is important for fine-tuning and learning new movements. However, it is not clear if the pattern of connections between modules in the brain is broadly the same across different animals from the same species, or if it varies between individuals. This is partly because it has not been possible to identify the same cells across different individual animals, and thus measure whether the connections are different.

Valera et al. used genetically engineered mice that produced a fluorescent marker in specific subsets of easily identifiable neurons in the cerebellum known as Purkinje cells. These groups of labeled Purkinje cells helped identify the boundaries of modules in the cerebellum. Valera et al. then stimulated another group of neurons, called the granule cells, that connect to Purkinje cells from several different modules. Measuring the resulting changes in the electrical activity of the Purkinje cells revealed the pattern of connections between these two types of neurons. These experiments showed that granule cells can connect Purkinje cells from distant modules and that neighbouring Purkinje cells often display similar connection patterns.

Valera et al. also found that Purkinje cells from different animals but at the same location in the cerebellum had similar patterns of connections with granule cells. This suggests that, in identified modules, there is little variability in connection patterns between individuals. However, the patterns of connectivity could be altered by processes related to learning, indicating that they can be organized in different ways and still work. Further experiments then showed that different individuals display consistent patterns of connectivity between granule cells and other major cell types in the cerebellum. This finding suggests that while the basic modules in the cerebellum are similar across different individuals, the connections between modules can be shaped by experience.

An important challenge for the future is now to understand how the connected modules combine the information they receive and drive the output of the cerebellum. This knowledge will help better describe how movements are controlled.

*2009*; *Kim et al., 2009*; *Xiao et al., 2014*; *Zhou et al., 2014*). Cerebellar modules are defined by the topographical organization of climbing fiber (CF) inputs to Purkinje cells (PCs), originating in the inferior olive, and PC to cerebellar nuclei connections. Olivo-cortical and cortico-nuclear projections delimit parasagittal zones and subzones of the cerebellar cortex called 'microzones' having similar CF receptive fields (*Oscarsson, 1979*; *Voogd and Glickstein, 1998*; *Sugihara and Shinoda, 2004*; *Voogd and Ruigrok, 2004*; *Buisseret-Delmas and Angaut, 1993*; *De Zeeuw et al., 2011*; *Voogd, 1967*). Thus, a microzone corresponds to the cortical element of a cerebellar module. PCs from a given microzone receive and integrate information from specific parts of the body and project to restricted areas of the cerebellar nuclei, which in turn send the cerebellar computation to the cerebral cortex, brainstem or spinal cord (*Voogd and Glickstein, 1998*). The complete olivo-cortico-nuclear loop defines the cerebellar module (Figure 1A; Figure 1—figure supplement 1; for review see *Ruigrok, 2011*). Mossy fiber (MF) inputs, the second major excitatory pathway of the cerebellum, partially overlap with the receptive fields of CFs when originating from the same body part (*Garwicz et al., 1998*; *Pijpers et al., 2006*; *Voogd et al., 2003*). Therefore, it has been suggested that the cerebellar cortex is subdivided into individual microzones each controlling a given set of muscles (*Apps and Garwicz, 2005*; *Thach et al., 1992*).

Some boundaries between different microzones can be identified by a family of neurochemical markers, the zebrins, selectively expressed in PCs (Figure 1—figure supplement 1) (*Apps and Hawkes, 2009*; *Brochu et al., 1990*). The level of expression of some of these markers has been associated with specific synaptic properties and intracellular transduction pathways at the parallel fiber (PF)-PC or CF-PC synapses (*Wadiche and Jahr, 2005*; *Paukert et al., 2010*; *Cerminara et al., 2015*; *Hawkes, 2014*) leading to the description of different intrinsic properties of PCs and different rules for plasticity induction between zebrin positive and negative bands. Combined with the regional differences in the cytoarchitecture of the cerebellar cortex (for review see *Cerminara et al.,*

*2015*), these findings demonstrate that the cerebellar cortex is composed of an ensemble of heterogeneous modules. Therefore, an appealing hypothesis would be that cell identity relies on its position in the cerebellar cortex and that specific communication rules between neighboring microzones might arise from this anatomical and neurochemical heterogeneity.

Intermodular communication might involve granule cells (GCs), which relay the MF input within the cerebellar cortex. GCs have long transverse axons, the PFs, that make glutamatergic synapses with hundreds of PCs across several microzones (*Harvey and Napper, 1991*; *Thach et al., 1992*). PFs could thus define an associative pathway that enables MF inputs from one original microzone to communicate with a distant one. Paradoxically, several studies have shown that PCs are excited by a unique set of GCs localized in the same microzone as themselves (*Cohen and Yarom, 1998*; *Bower and Woolston, 1983*; *Brown and Bower, 2001*; *Isope and Barbour, 2002*). These results suggest that local GCs provide the major input to PCs and that the majority of GC-PC synapses are silent (*Ekerot and Jorntell, 2001*; *Isope and Barbour, 2002*). In contrast, other studies demonstrated that PCs can be driven by a restricted subset of GCs belonging to a different microzone (*Ekerot and Jörntell, 2003*; *Dean et al., 2010*; *Jörntell and Ekerot, 2002*; *Ekerot and Jorntell, 2001*). Finally, recent studies have suggested that PCs can be excited by a broad range of GCs (*Walter et al., 2009*) and that feedforward inhibition through the MF-GC-molecular interneuron pathway might play a role in the restriction of receptive fields in PCs (*Cramer et al., 2013*; *Santamaria et al., 2007*). Finally, this wide range of functional connectivity might explain the regional differences in information processing (*Cramer et al., 2013*) underpinned by the molecular and cellular heterogeneity between individual microzones and by the specific topographical organization of MF and CF inputs in the cerebellar cortex.

We therefore investigated the functional synaptic organization of the excitatory MF-GC-PC pathway. In order to understand how information is processed in individual and adjacent microzones, we systematically mapped the functional synaptic organization between GC and PCs, molecular interneurons (MLIs) and Golgi cells (GoCs), targeting one set of identified cortical microzones in cerebellar modules of lobules III and IV of the anterior lobe. We describe here an activity-dependent stereotyped and predictable modular organization of GC inputs mediated by PFs, which pinpoints the operational rules of information processing in cerebellar modules.

## Results

### Targeting microzones in acute cerebellar slices

In order to address the hypothesis of specific functional synaptic communication between cerebellar microzones, we developed an experimental protocol that allowed us to target microzones in the medial parts of lobules III and IV, which receive inputs from the proximal hindlimb and forelimb. This region of the cortex is involved in the adaptive control of posture and locomotion; also, CF and MF inputs have already been well described in previous studies (*Ji and Hawkes, 1994*; *Voogd and Ruigrok, 2004*; *Sugihara and Shinoda, 2004*; *Garwicz et al., 1998*; *Andersson and Oscarsson, 1978*; *Matsushita et al., 1991*; *Matsushita and Tanami, 1987*; *Matsushita, 1988*; *Yaginuma and Matsushita, 1987*). Microzones are defined by their CF and cortico-nuclear projections (*Figure 1—figure supplement 1*) (*Andersson and Oscarsson, 1978*; *Oscarsson, 1979*; *Voogd and Ruigrok, 2004*). Since CF projections share common boundaries with the zebrin II pattern of expression in PCs (*Apps and Hawkes, 2009*; *Sugihara and Shinoda, 2004*), experiments were performed in the EAAT4-GFP strain of mice, in which zebrin II bands are defined by the expression of eGFP in PCs (*Figure 1—figure supplement 1*; Materials and methods) (*Gincel et al., 2007*). The use of two distinct specific antibodies against zebrin II/aldolase C (Materials and methods) confirmed that EAAT4-eGFP mice express GFP in zebrin II bands (*Figure 1—figure supplement 1*). By comparing EAAT4-eGFP/zebrin II expression and cortico-nuclear projections, we postulated that seven putative microzones extend to the P2$^+$ band on both sides of the midline (*Figure 1—figure supplement 1*).

So that the description of the cerebellar afferents could be completed, bilateral MF projections were reconstructed after the in vivo unilateral injection of AAV2/1-GFP viruses (N = 4 animals) or fluoro-ruby (N = 2 animals) into the external cuneate nucleus, which relays inputs from the forelimbs, or AAV2/1-GFP viruses into the lumbar spinal cord (L3–L5; N = 1 animal), which conveys inputs from the hindlimbs (*Figure 1B,C*; *Figure 1—figure supplement 2*; Materials and methods). Projections

from these two precerebellar nuclei were mutually exclusive in lobules III and IV, identifying an alternation of two to three parasagittal bands (mean width = 193 ± 24 μm, *Figure 1C,D*; n = 13 slices, N = 4 animals for cuneate projections; n = 4 slices, N = 1 animal for spinal projections) in the mediolateral axis. Intensity plot profiles of GFP labeling in the GC layer (Materials and methods) allowed us to determine the boundaries of individual MF bands (*Figure 1D*). Notably, when measuring the GFP level of expression, 22% of cuneate projections were found in the contralateral vermis, illustrating the importance of bilateral MF inputs. Interestingly, the comparison of the zebrin II pattern with MF

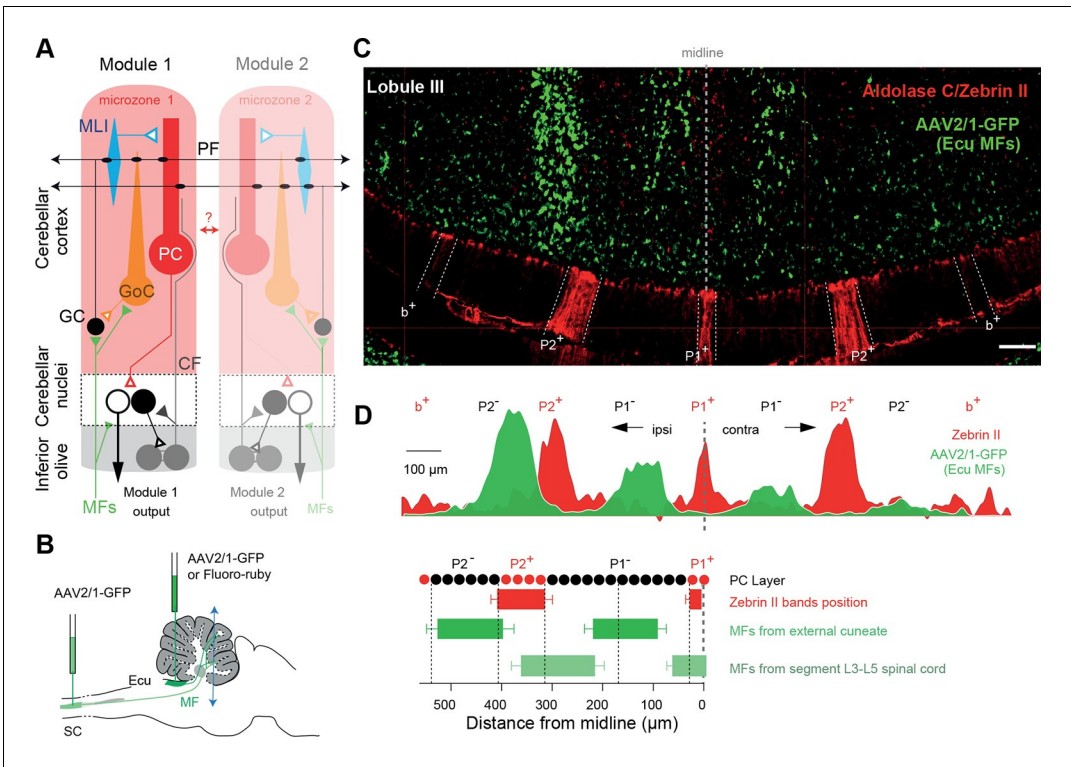

**Figure 1.** Identification of anatomical cortical microzones by the zebrin II bands (**A**) Diagram of two cerebellar modules. One module is composed of a cortical microzone (light red area), the target area of Purkinje cells (PC, red) in the cerebellar nuclei (black and white cells) and a group of olivary cells (gray) sending their climbing fibers (CF, gray line) to the cortical microzones and cerebellar nuclei. Mossy fibers (MFs, green lines) send sensorimotor information to the cerebellar nuclei and the microzones. The parallel fibers (PFs), the axon of granule cells (GCs), cross several microzones belonging to different modules. The red double arrow between the two modules illustrates intermodular communication. GoC: Golgi cell; MLI: molecular interneuron; filled triangles indicate excitatory synapses; empty triangles indicate inhibitory synapses. (**B**) Diagram illustrating AAV2/1-GFP or fluoro-ruby injection (green pipettes) sites in the external cuneate nucleus (ECu) and in segment L3–L5 of the spinal cord (SC). The blue double arrowhead line indicates the localization of the coronal section shown in panel C. (**C**) Coronal section across lobule III of the cerebellar cortex showing GFP fluorescence in MF rosettes following viral injection in the external cuneate nucleus (Ecu-MFs, green), aligned with anti-aldolase C/zebrin II (ZII, red) immunostaining. White dotted lines highlight positive zebrin II bands (P1+, P2− and b+ from the midline). (**D**) Upper panel, superimposed intensity plot profiles of the molecular layer (red) and granule cell layer (green) section shown in C illustrating bilateral MF projections. Upper labels, positive and negative zebrin bands. Black dotted line, midline. Lower panel, summary of MF inputs projection pattern in vermal lobule III from external cuneate (13 slices/4 animals) and spinal cord (4 slices/1 animal) compared to zebrin II bands (two positive bands: P1+, P2+ in red; and two negative bands: P1− and P2−).

The following figure supplements are available for figure 1:

**Figure supplement 1.** Zebrin bands as an accurate positioning tool.

**Figure supplement 2.** Mossy fiber projections from the external cuneate nucleus and spinal cord (segment L3-L5).

inputs identified overlapping parasagittal bands (*Figure 1D*) (*Ji and Hawkes, 1994*) with one shared boundary at the P2$^+$/P2$^-$ transition. Also, MF inputs subdivided the P1$^-$ band into smaller regions. Finally, CF and MF (*Figure 1D*) inputs delimit 13 regions having a unique combination of inputs between P2$^+_{ipsi}$ and P2$^+_{contra}$. These regions might therefore correspond to smaller processing units than microzones (*Ozden et al., 2009*; *Tsutsumi et al., 2015*). Using this anatomical description, we then targeted individual microzones in acute transverse cerebellar slices, and determined the spatial organization of GC excitatory inputs to PCs, GoCs and MLIs.

## Heterogeneous GC input maps to PCs revealed distant GC inputs

We choose to specifically study monosynaptic excitatory inputs from GCs to other cell types of the cerebellar cortex, since PFs can cross many microzones and communicate information over a long distance (*Pichitpornchai et al., 1994*; *Harvey and Napper, 1991*). Individual PCs (n = 49 cells, N = 18 animals) were whole-cell patch-clamped in P1$^+$, P1$^-$ and P2$^+$ zebrin II bands of EAAT4-GFP mice, and RuBi-glutamate was systematically uncaged in the GC layer using point-scan laser photostimulation (*Figure 2A,B*; width mapped = 664 µm centered on the recorded cell; Materials and methods) on both sides of the midline while inhibitory transmission was blocked. As uncaging led to a compound synaptic current in PCs initiated by trains of action potentials in GCs (*Figure 2B,C*; *Figure 2—figure supplement 1*; Materials and methods), the efficacy of GC patches in eliciting synaptic current in PCs was assessed by calculating the mean synaptic charge expressed as a Z-score (Materials and methods; *Figure 2—figure supplement 2*; *Figure 2C,D*). This representation allowed us to compare GC input maps between PCs with different background noise. Since MFs from the external cuneate and the spinal cord project to the entire depth of the granule cell layer (*Figure 1C*), glutamate uncaging sites at various depths of the GC layer but at the same mediolateral position were pooled and the evoked maximal synaptic input was considered. This maximal response was used as an indicator of the connectivity strength between the GCs and the PC along the mediolateral axis (*Figure 2B,D*). Interestingly, patterns of GC inputs in PCs were found to be highly heterogeneous (*Figure 2D*), with dense local inputs (mean local charge = 10.3 ± 13.8 pC, n = 49) and silent sites, that is, no GC patch had a Z-score > 3.09 at any depth of the GC layer (38% of sites were silent; Z-score < 3.09; Materials and methods). Remotely connected GC patches separated by silent sites were always found and were also frequently observed on contralateral sites (mean charge in distant patches = 6.1 ± 8.8 pC. Mean distance from the local peak = 215 ± 86 µm). In five cells out of 49, no obvious local inputs were observed although distal inputs were present. Altogether, these findings demonstrate that while local GCs make a strong connection with PCs most of the time, exceptions may occur. Also, GCs located at several hundred micrometers from the recorded PC can make strong connections via their PFs, while neighboring patches can be silent, indicating that information originating in distant microzones can have a strong influence on a given PC.

## Neighboring PCs share the same GC connectivity map

The connectivity map of one single PC might not account for the full population of PCs in a given microzone. Also, a single group of GCs is unlikely to influence a microzone by targeting a single PC. Therefore, if microzones communicate together via the PFs, PCs belonging to a given microzone should share several groups of functionally connected GCs from a distant microzone. We tested this hypothesis by recording pairs of neighboring PCs having the same zebrin identity (*Figure 3*) and determined their connectivity map. In the example shown in *Figure 3A,B*, strong GC inputs were observed from local GCs, in P2$^+$ ipsilateral and in P2$^+$ contralateral bands in both PCs, while no or few significant inputs originated in the P1$^-$ band. Although some strongly connected GC sites were specific to one PC of the pairs, we found a significant correlation between patterns of GC inputs for eight out of nine pairs of adjacent recorded PCs (*Figure 2C*; r = 0.74 ± 0.14, n = 9 pairs, N = 5 animals), indicating that in a given animal, PCs belonging to the same microzone and sharing similar MF inputs also displayed similar GC synaptic input maps. These findings suggest that neighboring PCs make powerful connections with a specific set of distant microzones while ignoring others.

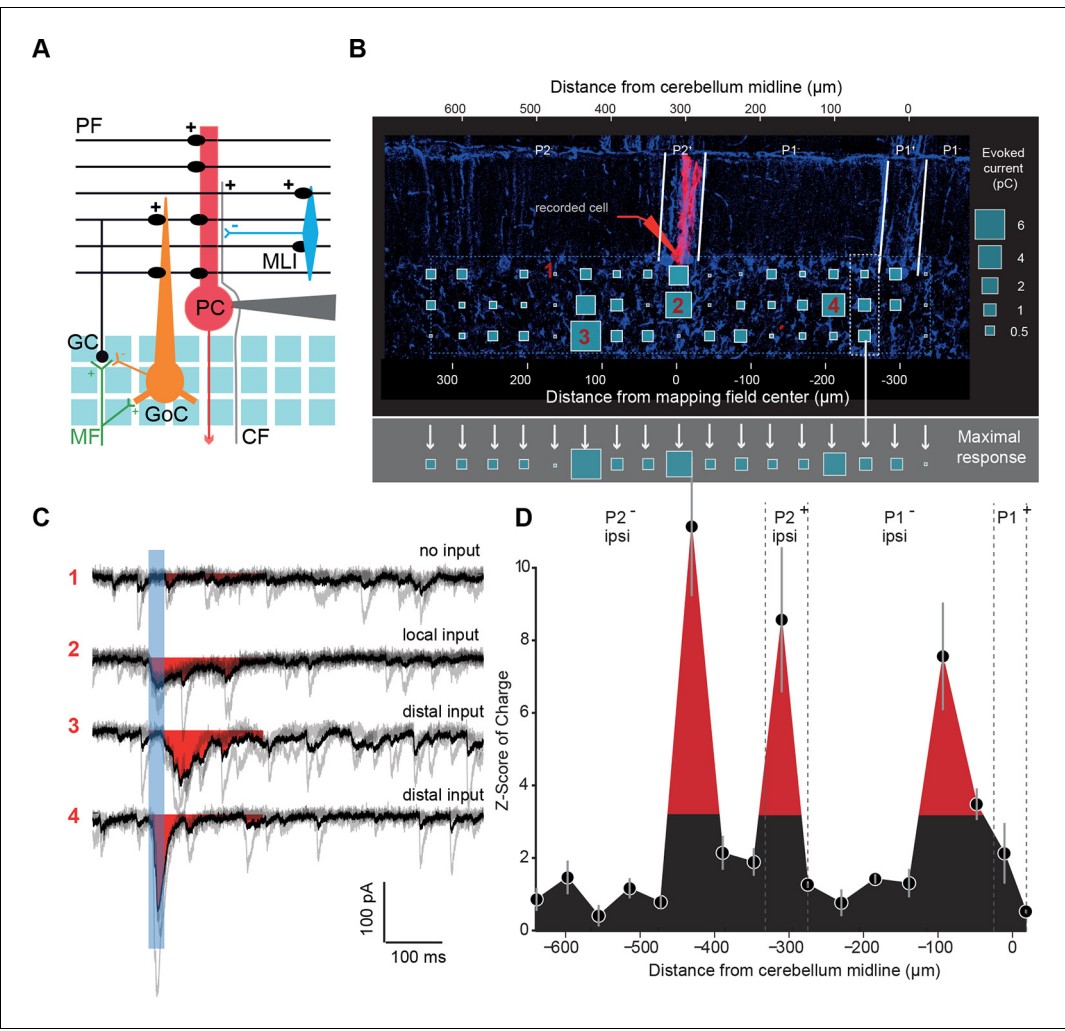

**Figure 2.** Granule cell input patterns to Purkinje cells reveal hotspots of connectivity. (**A**) Experimental design and simplified diagram of the cortical microcircuit. Purkinje cell (PC) synaptic inputs were recorded. Photorelease of RuBi-glutamate at multiple locations of the granule cell (GC) layer (blue squares) mimics GC (black) activation by mossy fibers (MFs; green). GCs contact PCs, Golgi cells (GoCs; orange) and molecular interneurons (MLIs; blue) along the mediolateral axis. Inhibition is blocked and climbing fibers (CFs) are not activated. PF: parallel fibers. (**B**) Example of a PC (red) recorded in an EAAT4-GFP acute cerebellar slice and filled with biocytin. The recorded cell was reconstructed and located using both GFP expression (not shown) and aldolase C immunolabeling (blue). The PC in this example is located in the P2$^+$ zebrin band. Blue squares indicate uncaging sites. The size of the square is proportional to the synaptic charge of the evoked current. Mediolateral response is given by the strongest response at all depths of the GC layer (i.e. maximal response in the white dotted box). Maximal responses along the mediolateral axis are reported in the gray area and define the connectivity pattern. The blue dashed box indicates the width of the photostimulation field. Please note the two scales: at the bottom is the distance from the recorded cell, while at the top is the distance to the midline. (**C**) Examples of evoked currents, from panel B. The blue bar indicates uncaging duration. The red area indicates the measured charge (window = 200 ms). (**D**) The connectivity pattern was expressed as the Z-score of charge, as a function of the distance to the cerebellar midline. The significance threshold was defined at Z = 3.09. Red areas are considered as functionally connected, while black areas indicate silent sites. Error bars illustrate the median from five mappings.

The following figure supplements are available for figure 2:

**Figure supplement 1.** Controls for photostimulation.

**Figure supplement 2.** Z-score representation of the granule cell input map in one recorded cell.

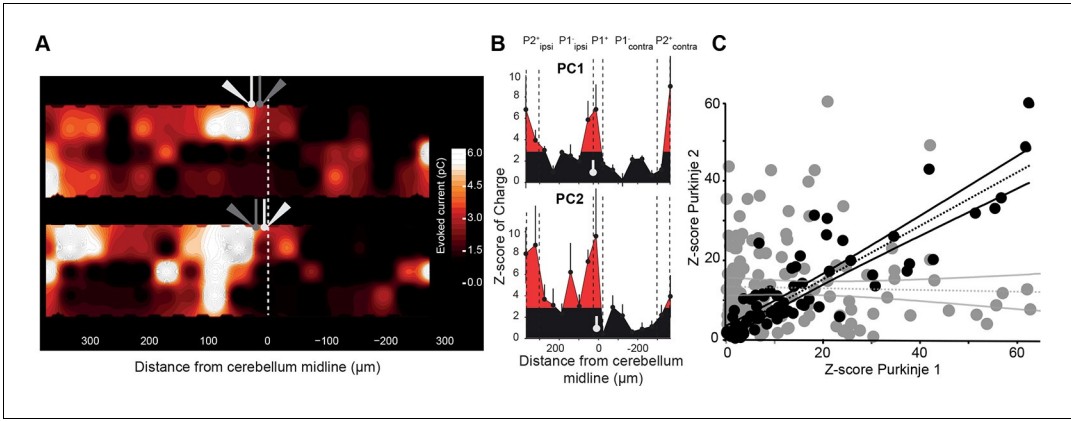

**Figure 3.** Neighboring Purkinje cells share similar granule cell input patterns. Two neighboring Purkinje cells (PCs) were simultaneously recorded and RuBi-glutamate was systematically uncaged. (**A**) Map of the recorded synaptic charge measured in PC1 (white cell, top) and PC2 (white cell, bottom). Most of the responding and silent sites were observed at the same location or close by, although a few differences can be observed. (**B**) Corresponding mediolateral granule cell (GC) connectivity pattern to PC1 (top panel) and PC2 (bottom panel) expressed as a Z-score of the synaptic charge. PC positions are indicated in white. (**C**) Site by site correlation of GC connectivity patterns between neighboring PC pairs (black dots, r = 0.74 ± 0.14; n = 8 pairs). Shuffled pairs showed no correlation (gray dots, r = 0.02 ± 0.09).

## Stereotyped GC input patterns across animals highlight specific links between identified regions

To investigate whether functional patterns of GC connectivity are specific to small groups of neighboring PCs in a given animal or are common features of the communication rules between cerebellar microzones, we set out to compare GC input patterns between groups of PCs belonging to different animals, but at the same location. PCs were grouped based on their position in the mediolateral axis between the P1[+] and P2[+] bands of lobule III/IV (group width = 100 µm; shift between two groups = 10 µm; N = 49 PCs from 18 mice; Materials and methods). In each PC group, the median pattern was determined (Materials and methods) and correlated with the median pattern of all the other groups of PCs. A correlation matrix was then built using the Pearson coefficient (r) as a marker of correlation between GC input patterns (*Figure 3A*; *Figure 4A*; Materials and methods). Strikingly, the correlation matrix revealed four clusters of PC groups displaying a highly correlated GC connectivity map (*Figure 4A*; clusters 1–4). Cluster identity was verified using a spectral co-clustering algorithm that simultaneously clusters rows and columns of a matrix without positional assumption (Materials and methods). The four clusters found were all centered on the diagonal line indicating that correlated PCs are neighbors. They are identified on the correlation matrix by dashed blue squares (*Figure 4A*). These results strongly argue that PCs from different animals but at the same specific location in the mediolateral axis (i.e. the vermal part of lobule III/IV) have selected similar GC sites. Notably, none of these clusters clearly matches with a set of anatomical boundaries as illustrated by the comparison with zebrin II bands and MF inputs from the external cuneate and the spinal cord (green rectangles, *Figure 4A*) indicating that boundaries of the functional organization between microzones differ from a pure anatomical description.

We then determined the median pattern of GC inputs for each of the four identified cluster of PCs (clusters 1, 2, 3 and 4 with n = 26, n = 14, n = 4 and n = 5 cells, respectively; *Figure 4B*). PCs defining cluster 1 displayed strong connections with local GCs and GCs located below ipsilateral and contralateral P2[+] but did not display any inputs from a large part of contralateral P1[−]. The P2[+]/P2[−] boundary also clearly identifies limits for cluster 2 that is characterized by weak local inputs. Also, in this group of PCs, a narrow contralateral band of GCs located 200 µm from the midline was always strongly connected. Contrasting with these two clusters, PCs that extend between 200 and 270 µm from the midline (cluster 3) were found to be broadly and strongly connected ipsilaterally by GCs spanning all zebrin positive and negative bands. Finally, PCs from the fourth cluster, located essentially in P2[+], were contacted by local GCs and GCs located in medial P1[−] and P2[−] bands. As

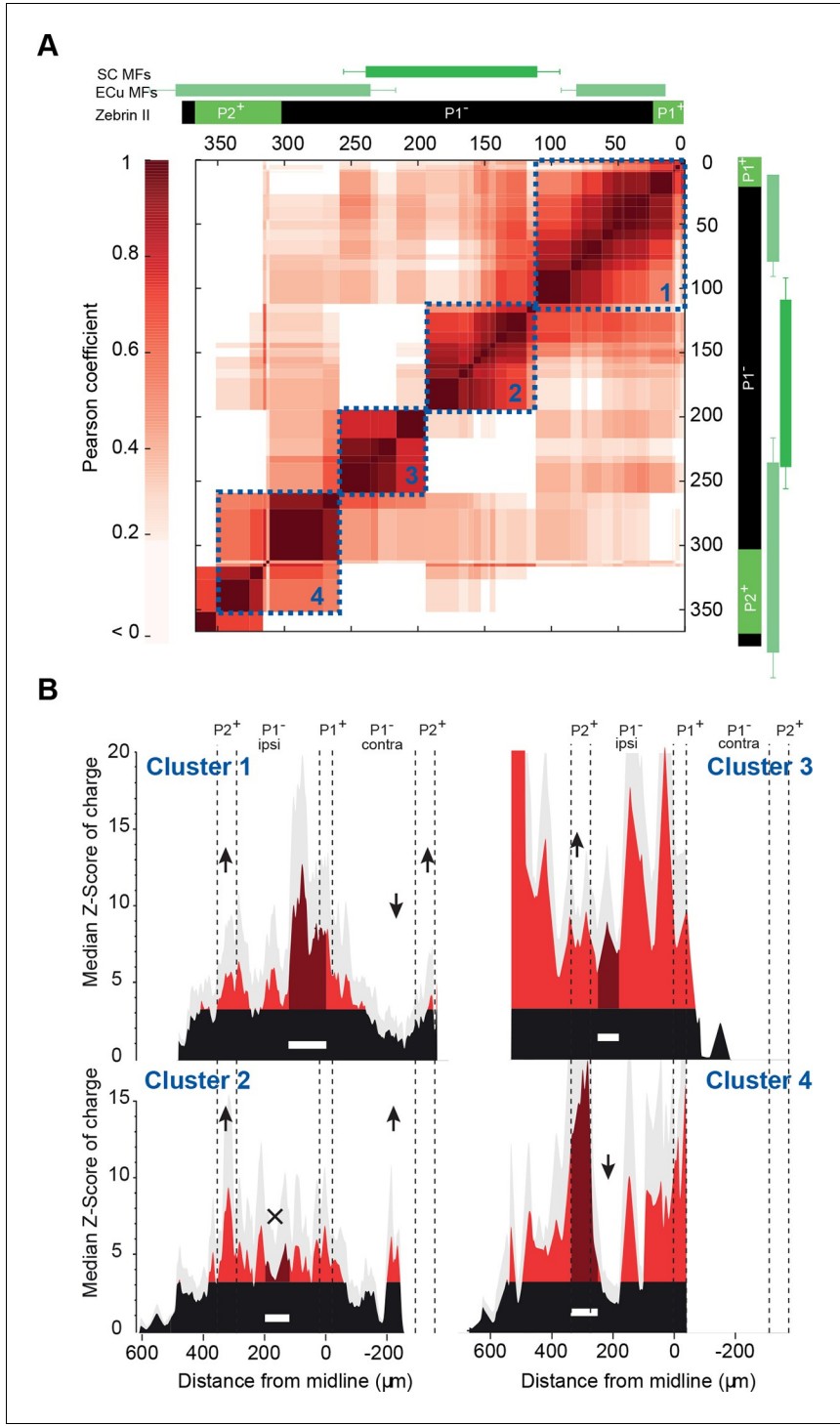

**Figure 4.** Granule cell input patterns to Purkinje cells from different animals define stereotyped clusters. (**A**) Correlation matrix of groups of neighboring Purkinje cells (PCs; N = 49 cells in 18 animals). Each group corresponds to the median granule cell (GC) input pattern of PCs located within a 100 µm window. Two consecutive groups are shifted by 10 µm. The x/y axes represent the center of each median pattern. All median patterns were compared to each other, and correlation was estimated using the Pearson coefficient. Four clusters of contiguous, correlated group of cells were identified using a co-clustering algorithm (blue dotted boxes, numbered 1 to 4 from the midline to P2$^+$). Zebrin II bands, external cuneate mossy fiber input (ECu MFs) and spinal cord MF input (SC MFs) locations are indicated in green, for comparison. (**B**) Median GC input patterns to the four clusters of PCs identified in A. The PC cluster position is represented as a white bar. Black: non-connected

*Figure 4 continued on next page*

*Figure 4 continued*

areas, Z-score < 3.09; red: significant connections, Z-score > 3.09. Error bars are in light gray. Local GC inputs (dark red) were usually observed, although weak for PCs belonging to cluster 2 (see black cross). Distal hotspots mentioned in the main text are indicated with upwards arrows. Note the systematic hotspot in P2$^+$. Silent regions mentioned in the main text are indicated with downwards arrows.

for the four identified clusters of correlated PCs, no systematic link between GC hotspots and anatomical boundaries has been observed. However, it should be noted that for all PC groups, strong GC connections were always found in ipsilateral P2$^+$ while a large area of the P1$^-$ band was silent. These findings showed that the highly heterogeneous patterns of GC connectivity are stereotyped and associated with four identified clusters of PCs, suggesting that a distinct functional organization linking distant regions might be superimposed on the anatomical microzonal framework.

## GC input patterns to GoCs and MLIs

In the cerebellar cortex, GCs also contact local interneurons, MLIs and GoCs, which respectively perform feedforward inhibition onto local PCs and feedback inhibition onto GCs. We therefore tested whether GC connectivity patterns to GoCs and MLIs displayed similar spatial functional properties. GC inputs to GoCs were examined using GlyT2-EAAT4-GFP mice (*Figure 5A*; Materials and methods). As opposed to PCs, patches of GCs connected to GoCs (n = 19 cells) were always found in the vicinity of the recorded cell. Within the mapped area, we found that 81% of GC synaptic inputs to GoCs were made by local GCs as assessed by the histogram of the median Z-score (*Figure 5A*), for GoCs recorded in any of the four clusters of PCs identified (*Figure 5—figure supplement 1*). Since dendritic gap junctions might have shunted and filtered GC inputs located in distant apical GoC dendrites (*Vervaeke et al., 2012*), potentially resulting in undetectable current at the soma, one subset of GoCs (n = 6) was recorded with the gap junction blocker carbenoxolone (100 µM) in the bath. No differences were observed in these mappings, and all cells were pooled. The pattern of GC inputs was centered on GoCs with a mean extension of 114.1 ± 70.1 µm, which falls within the average range of the reconstructed GoC axonal plexus (214 ± 30 µm, n = 10; inset *Figure 5A*). Neither the connectivity map nor the extension of the axonal plexus were limited by zebrin boundaries (*Figure 5—figure supplement 1*) as opposed to GoC apical dendrites (*Sillitoe et al., 2008*). Thus, unlike GC inputs to PCs, excitatory inputs to a given GoC appear to be mostly restricted to GCs that can be targeted by the local axon of the same GoC, confirming that they implement a local inhibitory feedback circuit in the GC layer (*Cesana et al., 2013*). Occasionally, small distal inputs were observed.

We then compared GC connectivity to MLIs (n = 7 cells, N = 4 animals) for cells recorded in the area of cluster 1 for PCs (*Figure 5B*). In MLIs, patterns of GC connectivity were measured using the number of events (expressed as a Z-score; Materials and methods) instead of the charge, as this measurement was less sensitive to background noise in this set of experiments. Similarly to GC-PC and GC-GoC connectivity, local GC inputs were identified. However, several patches of distant GC connections were also found in the histogram of the median Z-score of synaptic events (*Figure 5B*; local peak, 80 µm from midline, two ipsilateral peaks, 195 µm and 320 µm from midline, and one contralateral peak, 135 µm from midline). Strikingly, some hotspots did not overlap perfectly with patches of GCs that elicit strong inputs in PCs from cluster 1 (*Figure 5B*, see arrow and compare with PC from cluster 1 in *Figure 4*), indicating that for PCs and MLIs located in the same area, different groups of GC inputs can be selected. These findings suggest that GCs can relay MF information to PCs through two distinct pathways, the MF-GC-PC and MF-GC-MLI-PC pathways.

## Activity-dependent tuning of GC input maps

Since the GC connectivity maps to PCs, MLIs and GoCs strictly match neither with the topography of the CF inputs nor with the boundaries of the MF projections, these heterogeneous and stereotyped functional maps might arise from a combination of the topographical organization of MF and CF projections and an activity-dependent synapse selection through long-term plasticity (*long-term* depression [*LTD*] and *long-term potentiation* [LTP]). In a set of PCs from cluster 1, we therefore tested whether GC connectivity maps could be altered by a protocol known to induce plasticity,

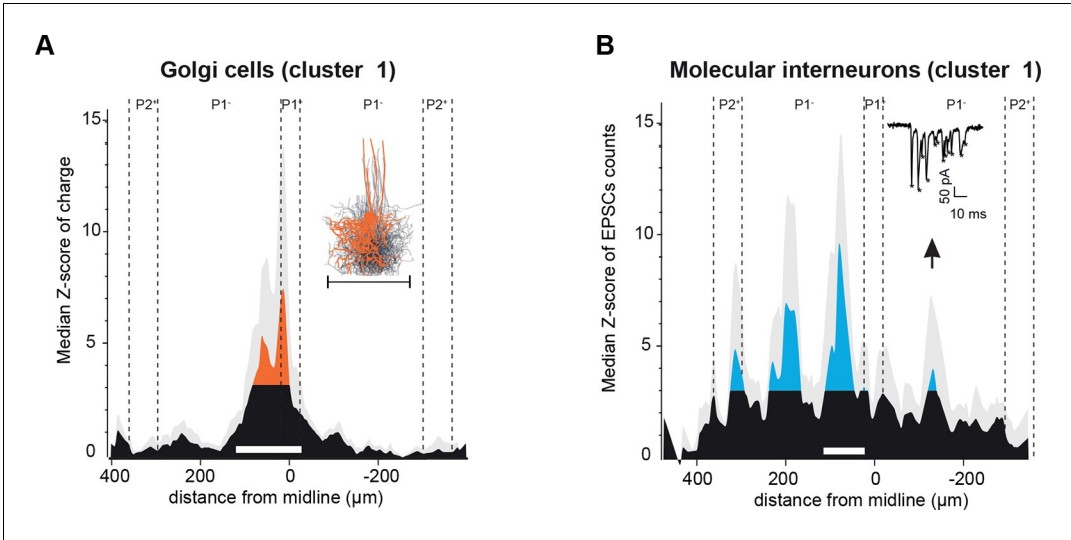

**Figure 5.** Spatial organization of granule cell (GC) input patterns to molecular interneurons and Golgi cells is distinct from GC input maps to Purkinje cells. (**A**) Median granule cell (GC) input pattern to Golgi cells (GoCs; Z-score of charge) for GoCs located at the same location as Purkinje cells (PCs) from cluster 1. Black: Z-score <3.09, orange: Z-score >3.09. Error bars are in light gray. Inset: overlay of all 3D-reconstructed GoCs, showing the extension of the axonal plexus, that is, the maximal region in which GoCs could inhibit GCs (same scale as the median pattern). (**B**) Median GC input pattern to molecular interneurons (MLIs; Z-score of number of *excitatory post-synaptic currents,* EPSCs) at the same position as PCs from cluster 1. The upward arrow indicates a hotspot of GCs contacting MLIs located in the cluster 1 region, but not PCs. Inset: example of EPSCs recorded following photostimulation. Stars indicate detected EPSCs.

The following figure supplement is available for figure 5:

**Figure supplement 1.** Spatial organization of granule cell input patterns to Golgi cells recorded in the area of cluster 2,3 and 4 for Purkinje cells.

---

either LTP or LTD (*Coesmans et al., 2004*; *Hartell, 1996*). After producing a first series of GC input maps (*Figure 6A*), we applied an electrical stimulation (1 Hz stimulation/5 min) to a large number of PFs in the molecular layer (mean evoked current in PCs = 1285 ± 500 pA, n = 12 which correspond to around 130 non-silent PFs; *Isope and Barbour, 2002*), and resumed the mapping procedure for at least 15 min. The initial averaged map was then compared site by site to the averaged updated map following stimulation (*Figure 6A-figure supplement 1*). After plasticity induction, 86 sites (22%; n = 8 cells) displayed a significant modification in synaptic charge ($\triangle$Z-score > 3.09 or < −3.09; *Figure 6A-figure supplement 1*; Materials and methods), while no effect was observed in the remaining sites. As already observed, this protocol induced postsynaptic LTP (green squares, *Figure 6A*; *Coesmans et al., 2004*) or LTD (blue squares, *Figure 6A*; *Hartell, 1996*) at the GC-PC synapse, indicating that we stimulated different PF beams converging on the recorded PC. All these changes were blocked by a combination of drugs that prevented the induction of plasticity (*Figure 6—figure supplement 1*). A negative correlation was observed (slope 0.31 and r = 0.38) between the initial synaptic weight of the GC site (high Z-score) and the sign of the effect after induction, with stronger connections leading to LTD while weaker connections undergo LTP (*Figure 6—figure supplement 1*). Indeed, selecting strong connections (Z-score > 6.5) and determining the averaged time course for all these sites led to a mean depressed charge of 27% after plasticity induction. Conversely, of the 56 sites (14% of total sites) that were potentiated, 33 were silent in the initial map (white 'x' in green squares in *Figure 6A*; *Figure 6—figure supplement 1*). This suggests that previously undetectable synaptic connections were awakened and that the overall connectivity map can be modified by activity. Indeed, we compared the histograms of the median Z-score of charge of this set of PCs before and after plasticity induction (*Figure 6B*). Although a few percent of the total number of PFs crossing the dendritic tree of the PCs had been stimulated, a new distant

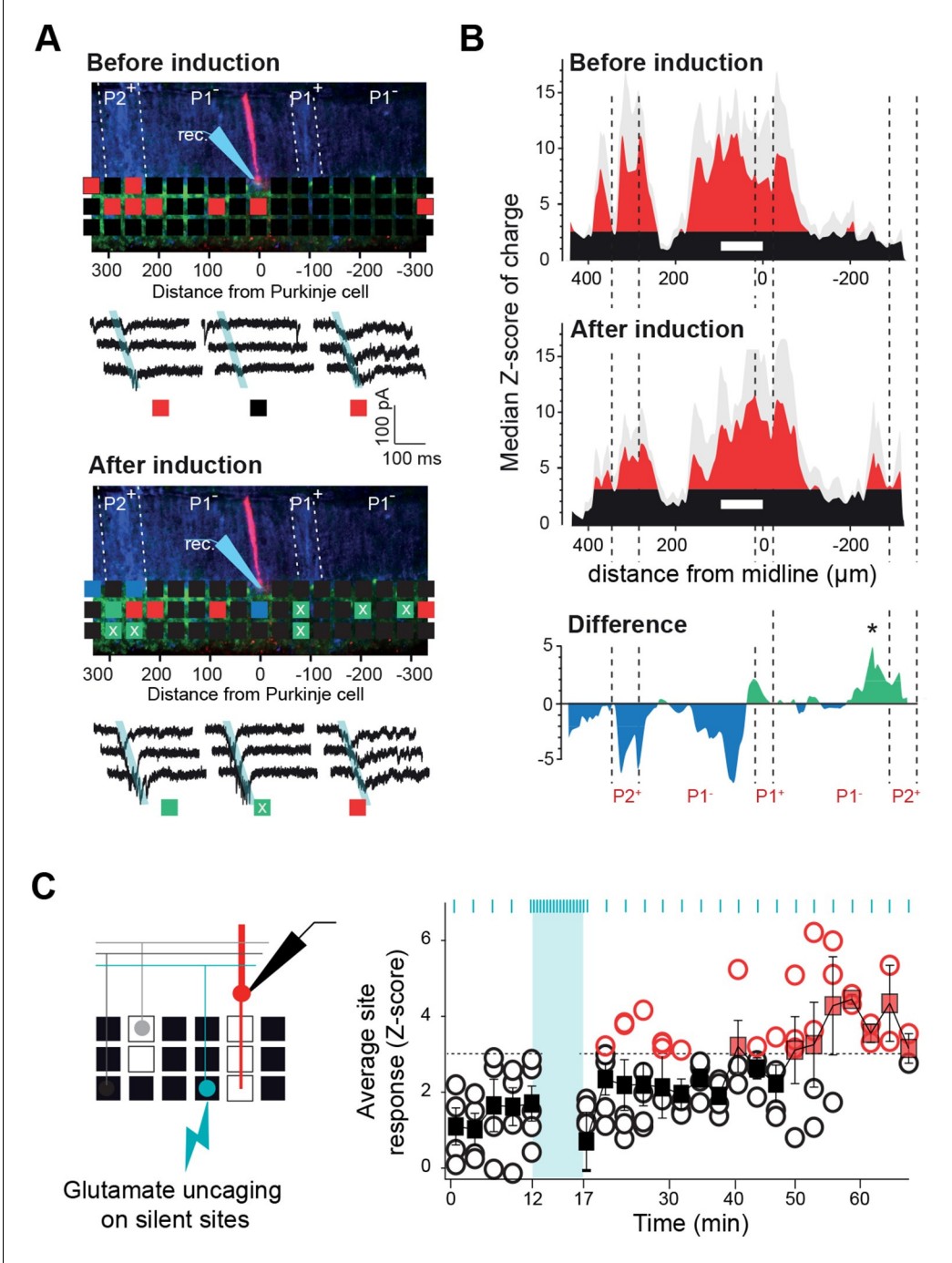

**Figure 6.** Tunable maps of granule cell inputs to Purkinje cells. (**A**) Photostimulation grid superimposed on a reconstructed slice before (top) and after (bottom) the long-term synaptic plasticity induction protocol (300 stimulations at 1 Hz). The stimulation pipette was positioned in the molecular layer far from the recorded Purkinje cell (PC). Zebrin II bands were confirmed by anti-aldolase C immunolabeling (blue) and are highlighted by a white dotted line. Recorded PCs are shown in red. Connected sites are in red, silent sites in black, depressed sites in blue, and potentiated sites in green. Awaken sites are indicated with a white cross. Traces illustrate examples of synaptic currents recorded in three sites showing different plasticity (same color code). (**B**) Median granule cell (GC) input patterns to PCs were computed before (top panel) and after the plasticity induction protocol (middle panel) for a group of PCs from cluster 1 (n = 7). A difference between GC input patterns (bottom panel) identified a newly connected region. (**C**) Silent sites can be specifically awakened. Silent sites were identified in an initial map and RuBi-glutamate was uncaged repetitively (one flash every 3 s for 5 min, blue bar) at these sites. Right panel,
*Figure 6 continued on next page*

*Figure 6 continued*
time course of evoked responses before and after induction (blue ticks indicates photostimulations). Individual
responses (n = 5 cells) are indicated as circles. Average responses are indicated by squares. Non-detectable
connections are in black, while significant connections are in red.
The following figure supplement is available for figure 6:

**Figure supplement 1.** Tunable maps of granule cell inputs to Purkinje cells.

region, initially silent, became significantly connected to the PCs belonging to cluster 1, demonstrating that the connectivity map is adjustable (see * in the panel 'Difference' in *Figure 6B*). Therefore, functional microzones may communicate through the selection of GC-PC synapses in a specific set of PC clusters.

In order to assess whether silent sites can be specifically awakened at any position in the mediolateral axis, the induction protocol was reproduced by uncaging glutamate specifically on silent patches of GCs (chosen at random distances from the recorded cell) at 1 Hz for 5 min (n = 5 cells; *Figure 6C*). In all cells tested, GC inputs became detectable after the induction protocol (mean △charge from −0.4 ± 0.33 to −1.27 ± 1 pC). These findings demonstrate that the functional pattern of GC inputs to PCs results from the active tuning of synapses.

## Discussion

Our data revealed that: (1) clusters of neighboring PCs share common rules for the selection of GC inputs (*Figure 7A*); and (2) PFs communicate MF information over a long distance in the cerebellar cortex linking cerebellar microzones through GC-PC and GC-MLI synapses while GC-GoC synapses appear mostly restricted to intra-microzonal communication (*Figure 7B*). Furthermore, we demonstrated that spatial patterns of connectivity are predictable and consistent between animals, suggesting that specific sets of cerebellar functional microzones are reproducibly combined together following similar operational rules. Finally, the functional synaptic organization of GC inputs to PCs and MLIs and GoCs does not strictly respect anatomical and neurochemical boundaries.

### Consistent and predictable communication between distant microzones

We identified highly heterogeneous patterns of GC excitatory inputs to PCs and MLIs highlighting hotspots of connected GCs in distant microzones. Hotspots were frequently observed at several hundred micrometers from the recorded PC or MLI. At first glance, these findings appear to be in contradiction with previous studies (*Harvey and Napper, 1991*; *Walter et al., 2009*). *Walter et al. (2009)* demonstrated that GC inputs are evenly distributed in the mediolateral axis although local connections were slightly stronger. However, as patterns of GC inputs were aligned on PC somas, positional information could not be considered in this study. Indeed, removing the positional information from our dataset reproduced these findings (*Figure 7—figure supplement 1*), indicating that the identification of the organization of GC input patterns is position specific. Therefore, whether we define a microzone by the topography of the CF inputs or the combination of CF and MF inputs (*Figure 1* and *Figure 1—figure supplement 1B*), our results strongly argue that PFs communicate excitatory MF information between distant microzones. Because inhibition was blocked in our experiments, we have to acknowledge that the GC input patterns of connectivity described here do not describe the net excitatory/inhibitory balance affecting PC dendrites. However, although no prediction about PC discharges can be made, the GC connectivity map onto MLIs observed in *Figure 5* suggests that PCs can be potentially inhibited by GCs from specific locations (*Figure 7*). Indeed, several studies demonstrated that feedforward inhibition participates in controlling the firing rate of PCs in individual microzones (*Dizon and Khodakhah, 2011*; *Santamaria et al., 2007*) and prevents the activation of distant PCs. Nonetheless, our data provide the first description of a conserved and predictable functional synaptic organization between one cortical layer – the granule cell layer – and the output stage of the cerebellar cortex – the Purkinje cell layer. The preferred organization of GC inputs to PCs and MLIs (in lobules III and IV of the anterior vermis) among different animals suggests that microzones may have specific roles in the context of the behavioral tasks

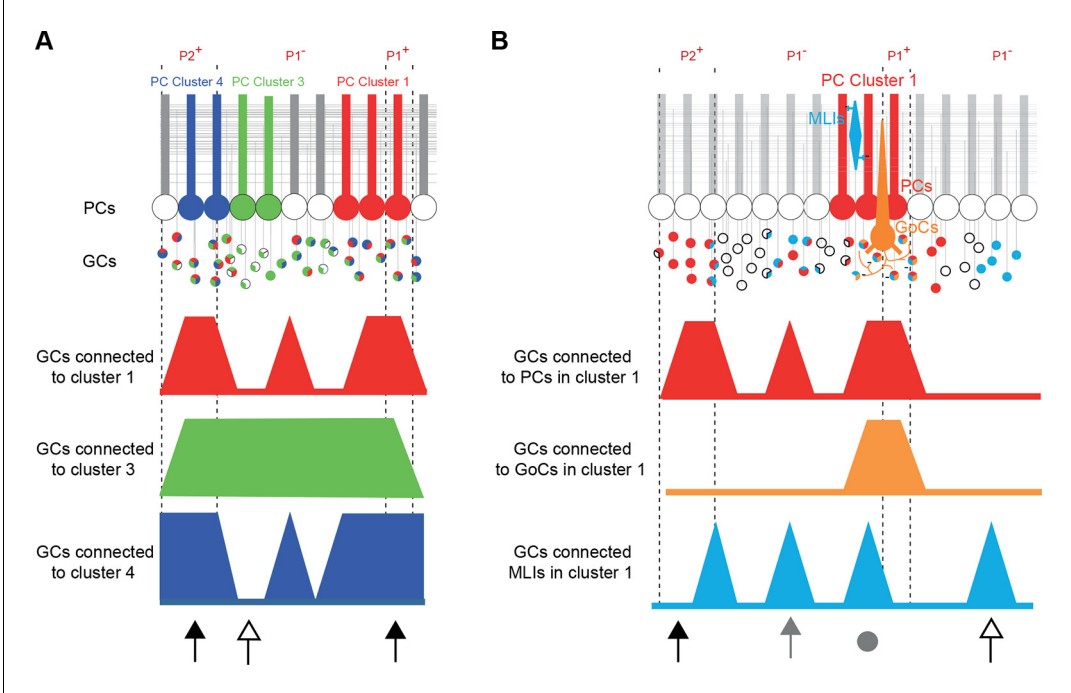

**Figure 7.** Summary. (**A**) Simplified diagram of granule cell (GC) input patterns to three Purkinje cell (PC) clusters. Zebrin bands are indicated with dotted lines. GC color indicates its postsynaptic target while white indicates either no target or an unknown target. Contiguous PCs display similar input patterns and each PC cluster presents a stereotyped input pattern. In the anterior vermis of lobule III/IV, some regions of the GC layer project to all recorded PCs (black arrows, for example in ipsilateral P2$^+$ or P1$^+$), while others can target a single PC cluster (white arrow). (**B**) Within a given microzone, the different cell types do not show the same GC input pattern. In cluster 1, PCs and molecular interneurons (MLIs) are sometimes contacted by GCs from the same region (either distal GCs, gray arrow, or local GCs, gray circle), while some groups of GCs contact either PCs (black arrow) or MLIs (white arrow). GoCs are mainly contacted by local inputs (gray circle) and implement local feedback inhibition.

The following figure supplement is available for figure 7:

**Figure supplement 1.** Mean histogram of normalized amplitude of granule cell inputs aligned on PC soma (see Discussion).

associated with this cerebellar region, notably during adaptive locomotion (*Apps and Garwicz, 2005*; *Dean et al., 2010*).

Our results are in agreement with in vivo studies demonstrating that PCs are activated by a restricted group of distant GCs (*Ekerot and Jörntell, 2001*; *2003*), although in these studies, local GC inputs were never observed. There are several possible explanations for this discrepancy: (1) the different areas of the cerebellum studied (i.e. the C3 zone rather than medial lobules III and IV) might have specific operational rules. Indeed, other groups studying other regions of the cerebellar cortex have demonstrated that local GCs provide the majority of inputs for the underlying PCs (*Bower and Woolston, 1983*; *Cohen and Yarom, 1998*; *Isope and Barbour, 2002*), for example in Crus I and II. Furthermore, in a recent in vivo study, while restricted patches of PC activation were observed in Crus II following forelimb stimulation, beam-like PC excitation elicited by PFs from distant GC inputs were identified in Crus I (*Cramer et al., 2013*). (2) In vivo, the receptive fields were defined by modulation of the PC firing rate. This parameter could have been misleading since a null net effect could still have masked correlated modifications of excitatory and inhibitory inputs. Notably, local GCs activate both PCs and MLIs in our experiments, suggesting that the resulting PC discharge is difficult to predict. In vivo, excitation or inhibition could be favored depending on the subset of GCs activated in a given context.

## Tunable patterns of GC inputs

Because similar patterns of GC inputs to PCs, GoCs and MLIs were found in the different animals tested, a major issue was to assess whether this organization was the result of a genetically encoded developmental program or if it could be modified by activity. We demonstrated that PF stimulation could induce either LTD at individual sites (*Hartell, 1996*) or LTP at individual sites (*Coesmans et al., 2004*) indicating that activity-dependent processes can tune GC input maps. Such variability in the sign of the plasticity is due to the fact that the stimulation pipette sampled a limited set of PFs constituting a small proportion of the total number of PFs in the lobule, but belonging to many different GC patches. This explains why only 22% of GC patches were affected by plasticity. At the center of the stimulated PF beam, focal synaptic inputs to PC dendrites have likely evoked large calcium transients, similar to CF inputs (*Hartell, 1996*), and thus induced LTD at these specific GC-PCs synapses. Focally evoked calcium transients could have been amplified at GC-PC synapses with a high synaptic weight, as stronger depolarization would have been elicited. Although focal stimulation in the molecular layer may appear non-physiological, a recent study demonstrated in vivo that sensory inputs can lead to clustered activation of PFs (*Wilms and Häusser, 2015*). At the periphery of the stimulation beam, sparse activation of PFs might have induced moderate calcium transients leading to LTP (*Coesmans et al., 2004*). Interestingly, new connections appeared systematically when we used a plasticity-inducing protocol at silent sites, demonstrating that quiescent synapses can be awakened following GC activation. These findings explain the appearance of a new peak of connected GCs in the median histogram of the Z-score of charges after plasticity induction. Indeed, plasticity occurred at almost every position in the mediolateral axis (see difference in *Figure 6B*), but at silent sites a strong bias toward LTP was observed (*Figure 6—figure supplement 1*). Because of the high efficiency of the stimulation protocol to awaken silent synapses (*Figure 6B*), a strong alteration of the GC connectivity map was likely to occur. Further experiments using specific MF stimulation will be required to assess whether such modifications in GC input maps are observed in physiological conditions. However, our results clearly demonstrate that the heterogeneity of the spatial map is an active process. These results are in agreement with previous in vivo studies (*Jörntell and Ekerot, 2002*; *Schonewille et al., 2010*; *Gao et al., 2012*; *Badura et al., 2013*), suggesting that distant microzones are associated through PFs by LTP mechanisms. LTP could favor information transfer between a set of GCs from one microzone and a group of PCs from another microzone. Conversely, LTD, which is physiologically induced by the conjunction of PF and CF inputs to PCs, could limit the communication between these microzones.

## The functional units process information from different origins

We have demonstrated here that PFs can link distant cortical regions that do not always match with known zebrin band markers or MF inputs. Notably, neither the PC clusters nor the GC spatial patterns of connections strictly match with the topography of the spinocerebellar or cuneate MF inputs carrying information from the hindlimb and forelimb, indicating that GC patches cannot simply be identified by the anatomical organization of a single source of MFs, and might combine MF information from different sources, as suggested in two recent studies (*Chabrol et al., 2015*; *Huang et al., 2013*; but see also *Bengtsson and Jorntell, 2009*). Discrepancies between anatomical and functional boundaries have also been shown using in vivo calcium imaging of CF signaling in PCs following sensory stimulation (*Tsutsumi et al., 2015*). Since $P2^+$ bands receive specific MF inputs from the hindlimb (*Jörntell et al., 2000*), the functional organization observed in our experiments might illustrate the importance of connecting cortical microzones controlling similar muscles in both limbs.

The term microzone was originally associated with both the description of the fine anatomical CF projection and the concept of unitary functional unit in the cerebellar cortex (*Oscarsson, 1979*). We described here clusters of neighboring PCs and MLIs having similar GC input connectivity maps that might represent individual functional units. However, they differ from the original *anatomical microzone* definition as they can span multiple zebrin bands. Therefore, a *functional microzone* likely computes information from different sources of MF inputs and frequently belongs to two adjacent zebrin bands as also proposed by *(Graham and Wylie, 2012)*. Since zebrin bands also define an array of biochemical markers that can have an important role in synaptic transmission, signal integration and plasticity (for example EAAT4 transporter, PLCbeta3 and 4, IP3R1 and mGluR1a receptors follow a specific zebrin pattern of expression) (*Cerminara et al., 2015*; *Hawkes, 2014*; *Paukert et al., 2010*;

*Wadiche and Jahr, 2005*), a functional microzonal unit might then combine several toolboxes in order to process incoming information. In light of the stereotyped organization of GC input patterns to PCs and MLIs, and the local restriction of the GC inputs to GoCs, we postulate that the area between $P2^+$ ipsi- and contralateral bands could be composed of 10–13 functional units with a mean extension of 70–100 µm, as already suggested by in vivo imaging studies (*Ozden et al., 2009*; *Tsutsumi et al., 2015*; *Schultz et al., 2009*). In the posterior lobe, zebrin bands and cerebellar inputs have a different topographical organization, notably in the vestibular lobules (*Voogd, 2011*), but a modular system has also been described suggesting that the communication between microzones through PFs is a general mechanism involved in information processing in the cerebellum. However, whether conserved and stereotyped patterns across animals is the rule in the posterior lobe needs to be demonstrated. In Crus I/II, animal-specific processing linked to vibrissa tactile discrimination might lead to unique communication rules, as sensory discrimination relies on individual history.

In conclusion, we demonstrated that cortical functional microzones of the cerebellar cortex communicate via the PFs and underlie an important level of coordination between cerebellar modules. These properties will favor the synchronization of PCs within specific modules and improve information readout by nuclear cells (*Person and Raman, 2012a*; *2012b*).

## Materials and methods

All experimental procedures conformed to French ministry and local ethics committee (CREMEAS) guidelines on animal experimentation.

### Tracing experiments

MF projections from the cuneocerebellar and dorsal spinocerebellar tracts were labeled by in vivo injection of viruses or fluoro-ruby into the external cuneate nucleus (N = 6) and the lumbar region of the spinal cord (N = 1), respectively. Male CD1 mice (P28–P35) were injected with recombinant adeno-associated viral particles (rAAV2/1, $2.8 \times 10^{12}$ GU/ml, N = 5) carrying cDNA for GFP expression under the CMV promoter, or with dextran tetramethylrhodamin (10 000 MW, fluoro-ruby, N = 2). Mice were anesthetized with an intraperitoneal injection of a mixture of ketamine (100 mg/kg), medetomidine (1 mg/kg) and acepromazine (3 mg/kg). For external cuneate targeting, stereotaxic injections were performed using the coordinates AP: −7.4 mm; Lat: 1.4 mm; DV: 3.3 mm from bregma. The virus or the dextran was loaded into a graduated pipette equipped with a piston for manual injections (Wiretrol II, Drummond Scientific Company, Broomall, USA). By applying gentle pressure, final volumes of 1.2 µl and 0.75 µl were delivered into the spinal cord and the external/cuneate nucleus, respectively, at an approximate speed of 250 nl/min. The pipette was left in place for at least 10 min after injection for virus diffusion. Spinal injections were achieved by inserting the pipette between adjacent vertebrae in the lumbar region. After 2–11 weeks of recovery, injected mice were sacrificed by transcardiac perfusion of paraformaldehyde 4% and cerebellar slices (50 µm) were prepared for subsequent immunohistochemistry and analysis. Zebrin bands were identified using a monoclonal (1/100 dilution; gift of Richard Hawkes, Calgary) (*Brochu et al., 1990*) or polyclonal (1/50 dilution; gift from Izumi Sugihara, Japan) (*Sugihara and Shinoda, 2004*) antibody against aldolase C. Intensity plot profiles were generated after confocal imaging in individual slices by measuring the intensity of the MF signal in the granular layer and the intensity of aldolase C/zebrin II labeling in the molecular layer of lobule IV. Measurements were performed using ImageJ software. For group data, MF intensity plot profiles were aligned to the $P1^+$ band (indicating the midline).

In four out of six animals injected in the external cuneate nucleus, the injection site extended into the entire lateral and dorso-ventral axes and 350–400 µm into the antero-posterior axis (*Figure 1—figure supplement 2*) of the nucleus. Cortical projections in lobules III and IV were measured in 13 different slices in these four animals. Injection in the lumbar segments of the spinal cord (L3–L5; 1.2 µl) resulted in an infected area of ≈1 cm in length centered around L3–L5 (see *Figure 1—figure supplement 2*). Cortical projections were measured in four different cerebellar slices.

## Slice preparation

Slices were prepared from P17–P90 male CD1 EAAT4-GFP or EAAT4-GlyT2-GFP mice. This strain was obtained by crossing EAAT4-GFP mice with a line expressing GlyT2-GFP (gift from H.U. Zeilhofer) (*Zeilhofer et al., 2005*), in which some GoCs express GFP. Mice were anesthetized by inhalation of isoflurane, and then killed by decapitation. Transverse slices were prepared as previously described (*Valera et al., 2012*). The cerebellum was dissected out and placed in cold artificial cerebrospinal fluid (ACSF) bubbled with carbogen (95% $O_2$, 5% $CO_2$), containing (in mM): NaCl 120, KCl 3, NaHCO$_3$ 26, NaH$_2$PO$_4$ 1.25, CaCl$_2$ 2.5, MgCl$_2$ 2, glucose 10 and minocycline 0.00005 (Sigma-Aldrich, USA). Then 300 µm-thick transverse slices were prepared (Microm HM 650V, Microm, Germany) in potassium-based medium, containing (in mM): K-gluconate 130, KCl 14.6, EGTA 2, HEPES 20, glucose 25, minocycline 0.00005 and D-AP5 0.05 (Sigma-Aldrich). After cutting, slices were soaked in a sucrose-based medium at 34°C, containing (in mM): sucrose 230, KCl 2.5, NaHCO$_3$ 26, NaH$_2$PO$_4$ 1.25, glucose 25, CaCl$_2$ 0.8, MgCl$_2$ 8, and minocyclin 0.00005 (Sigma-Aldrich) and maintained in a water bath at 34°C in bubbled ACSF.

## Electrophysiological recording

Whole-cell patch-clamp recordings in voltage-clamp mode were obtained using a Multiclamp 700B amplifier (Molecular Devices, USA) and acquired with WinWCP 4.2.x freeware (John Dempster, SIPBS, University of Strathclyde, UK). Pipette resistance was 3–4 MΩ for PCs, 6–8 MΩ for GoCs and 10 MΩ for MLIs. Series resistance was monitored and compensated (80%–90% typically) in all experiments, and cells were held at −60 mV. The internal pipette solution contained (in mM): CsMeSO$_4$ 135, NaCl 6, HEPES 10, MgATP 4 and Na$_2$GTP 0.4. pH was adjusted to 7.3 with KOH and osmolarity was set at 300 mOsm. Biocytin (Sigma Aldrich) and neurobiotin (Vector Laboratories, USA) were added (1 mg/ml each) for cell reconstruction. Voltages were not corrected for the liquid junction potential, which was calculated to be 9 mV (i.e. the membrane potential was 9 mV more hyperpolarized than reported). We accepted recordings for which the inward current at −60 mV did not exceed 1 nA for PCs and 250 pA for other cells. Synaptic currents in PCs were low-pass filtered at 2 kHz, then sampled at 20–50 kHz. All recorded cells were located in lobule III or IV. All experiments were performed at 34°C using the same bubbled ACSF. We blocked inhibitory transmission and NMDA, adenosine, CB1, GABA$_B$ and mGluR1 receptors to limit the modulation of *excitatory post-synaptic current* (EPSC) amplitude by activity-dependent activation of these receptors. They were respectively blocked using (in mM): picrotoxin 0.1, strychnine 0.001, D-AP5 0.05 (Ascent Scientific, Abcam Inc), DPCPX 0.0005, AM251 0.001, CGP52432 0.001 and JNJ16259685 0.002 (Tocris-Cookson, UK).

An initial set of mappings was obtained during plasticity induction protocols. A glass pipette was positioned in the upper half of the molecular layer, 500 µm away from the recorded PC, and 300 stimulations were performed at 1 Hz in current-clamp mode. After a resting period of 5 min, photostimulation was resumed and the entire mapping was repeated at least four times. Perfusion contained picrotoxin, strychnine, CGP52432 and AM251 at the concentrations described above. For plasticity experiment in *Figure 6C*, the following internal solution was used (in mM): K-gluconate 136, KCl 4, MgCl$_2$ 1, HEPES 10, Na$_2$ATP 4, NaGTP 0.4, sucrose 16, pH 7.3, and inhibition was blocked with bicuculine (20 µM).

## Photostimulation

Uncaging experiments were performed using bath-applied RuBi-glutamate (100 µM, Ascent Scientific), as described by *Fino et al., 2009*. We used the point-scan mode of a confocal microscope (FV300, Olympus, Japan) for illumination, mounted with a diode-pumped solid-state blue laser (20 ms and 30 mW laser pulses at 473 nm [CrystaLaser, USA], through a 20× objective [Olympus, Japan]). The beam of blue light (473 nm) was controlled by a set of mirror galvanometers and focused using a low-aperture objective (20×/0.5 NA). This optical arrangement was designed to generate an almost cylindrical beam that could penetrate the slice, as assessed by the constant synaptic charge recorded when the focal plane of the photostimulation was increased to a depth of 100 µm in the slice (*Figure 2—figure supplement 1D*). Mapping was carried out using a software-based point-scan stimulation (PAPP, Fluoview 300) at identified positions, with a step of 41.5 µm every 3.5 s. The stability of the photostimulated current was assessed (*Figure 2—figure supplement 1G*).

In each experiment, the recorded cell was placed at the center of the field and the slice was positioned with the PC layer aligned with the X-axis of the grid. Grid extension was 332 µm on each side of the recorded cell above the GC layer. A range of 34–85 sites was sampled several times for each cell (mean number of mappings/cell = 5.1 ± 2.3). No sites were stimulated with frequencies higher than 0.008 Hz.

## Controls for photostimulation

RuBi-glutamate uncaging was used to study the functional excitatory synaptic connectivity of the cerebellar cortex. This compound was chosen because it incurs fewer non-specific effects than MNI-glutamate (*Fino et al., 2009*) during quantitative laser-scanning photostimulation (*Shepherd et al., 2005*). Since these caged compounds partially block GABAergic currents, GC inhibitory transmission was blocked.

The diffusion of uncaged RuBi-glutamate during laser stimulation was estimated by recording currents evoked in PC dendrites in the horizontal plane in 20 µm steps(*Figure 2—figure supplement 1A*). Action potentials were blocked with tetrodotoxin (TTX, 1 µM). Because of the thickness of the PC dendritic tree itself, direct stimulation of glutamate receptors yielded an upper limit for the size of the spot (half-width = 33.0 ± 1.8 µm, n = 9). Since the lateral extension of GC dendrites could increase the responsive area, we also estimated this extent by recording GCs both in loose cell-attached mode (measuring evoked action potentials, the half-width of evoked action potentials was 45.5 ± 10.9 µm, n = 7) and in whole-cell mode (measuring evoked currents, the half-width of direct stimulation was 59.6 ± 2.1 µm, n = 7) while uncaging glutamate in10 µm steps (*Figure 2—figure supplement 1B,C*). The average maximum number of evoked action potentials at the center of the spot was 19.6 ± 6.6 (n = 7) indicating that even GC-PC connections with a very low probability of release could evoke a current to the recorded PCs (*Isope and Barbour, 2002*). The distance between two photostimulated sites (41.5 µm) was chosen to ensure that neighboring sites were almost independent, and that all parts of the granular layer was sampled (*Figure 2—figure supplement 1*). We estimated that each beam could illuminate several hundred GCs per 10 µm of slice thickness. A burst of action potentials was triggered in this population (*Figure 2—figure supplement 1C*), where glutamate uncaging elicited currents of up to 10 pC. Since the synaptic weight of a single GC-PC connection is approximately 0.1 pC (*Isope and Barbour, 2002*), tens of active GC connections (depending on the depth of the GC layer in the slice) could be stimulated simultaneously, attesting to the efficacy of the photostimulation protocol.

We estimated the impact of the focal plane on GC responses. GC inputs were recorded in PCs while the focal plane was changed in 10 µm steps. The average current was not significantly different between focal planes (*Figure 2—figure supplement 1D*).

Since the detection of distal connected GCs could be affected by the inclination of the slice, we used GFP-positive Lugaro cell axons, which run parallel to the PF, from EAAT4-GlyT2-GFP slices and determined granule cell layer slope (*Figure 2—figure supplement 1F*). A total of 130 axons were traced in 13 slices after the experiment, and the mean deviation from the horizontal axis was 0.85 ± 0.82 deg, suggesting that the absence of distal responses was not due to a tilt in the slice.

## Analysis of synaptic charge

Measurements and analyses were performed using Python software written in-house and open source software, OpenElectrophy (http://neuralensemble.org/OpenElectrophy/) (*Garcia and Four-caud-Trocmé, 2009*). Data were stored in SQL databases. Synaptic charges evoked by photostimulation of granule cells were detected during a window of 0–200 ms after illumination*Figure 2—figure supplement 2*. The standard deviation of the background noise was detected as follows: peak amplitude was detected using a 200 ms window at the end of each trace, then the histogram of noise was built and its standard deviation was determined. The Z-score of synaptic charges for each site was calculated using the following equation: $Z\text{-score}_{site}$ = (mean synaptic charge at one site − mean charge of the noise)/(standard deviation of the noise). A Z-score of 3.09, corresponding to a significance level of 0.001, was chosen to define significant and silent sites. This conservative choice of threshold was chosen in agreement with visual inspection of the traces, to ensure that any evoked current was not simply due to spontaneous activity.

Although glutamate uncaging in the GC layer did not evoke direct currents in PCs, the direct stimulation of GoC somata or basolateral dendrites occurred at some sites. The slow direct current was filtered using a median filter applied to each individual trace. Evoked synaptic currents are much faster and were slightly affected. The MLI Z-score was determined using the number of individual EPSCs evoked by glutamate uncaging in a time window of 200 ms after uncaging compared to the number of individual EPSCs in a distant time window.

### Reconstruction of GC input patterns and correlation matrix

After the recordings, slices were fixed in 4% paraformaldehyde and zebrin bands were identified using a monoclonal (gift from Richard Hawkes, Calgary) (*Brochu et al., 1990*) or polyclonal (Izumi Sugihara, Japan) (*Sugihara and Shinoda, 2004*) antibody against aldolase C. Biocytin/neurobiotin-filled cells were visualized with a streptavidin Alexa Fluor conjugate (Thermo Fisher Scientific, USA). In order to compared reconstructed cell positions between animals, the average width of zebrin bands in lobules III and IV was determined (average $P1^+$ width = 48.5 ± 13.6 μm, average $P1^-$ width = 275.6 ± 70.3 μm, average $P2^+$ width = 64.6 ± 19.3 μm and average $P2^-$ width = 445.2 ± 60.5 μm, n = 89) and cells were positioned within these coordinates. A 3D reconstruction of GoCs was obtained using the free software Vaa3d (*Peng et al., 2014*). In each mediolateral position of the GC layer, the site with the maximum Z-score value was used to build spatial patterns (*Figure 2B,C*). For paired correlation and median Z-score histograms, spatial patterns were convolved, using a triangular kernel (half-width = 18 μm).

The correlation matrix was built as follows: the representative pattern of GC inputs at each position was given by the median Z-score of synaptic charges for a group of PCs in a range of 100 μm. A full set of median patterns was generated by shifting the averaging window by 10 μm. The Pearson correlation was calculated between each couple of median patterns, and the reported location in *Figure 4A* corresponds to the center of the median pattern. Biclustering was performed using spectral co-clustering from the sklearn Python library (*Abraham et al., 2014*). In brief, the algorithm rearranges the rows and the columns of the correlation matrix to make biclusters contiguous without assumption. If the order of rows and columns is not changed, it indicates that the initial data order already presented a clustered organization.

### Statistics

Means are reported with standard deviations, while error bars in figures represent SEMs or MADs (median absolute deviation) in cases where the mean or the median, respectively, was used. Unless otherwise stated, statistical tests used were the non-parametric Mann-Whitney U and Spearman rank order tests. Correlations were calculated using the Pearson coefficient.

## Acknowledgements

We thank Thomas J Younts, N Alex Cayco Gajic, Chiara Baragli, Clément Léna and Angus Silver for their comments, Samuel Garcia for his technical assistance with Python programming and Nicolas Polin (CeStats, University of Strasbourg) for his assistance with statistical methods. We are grateful to Pascale Koebbel, the Mouse Clinical Institute, Sophie Reibel-Foisset and Laurence Huck for their technical assistance. We thank S Rasika (Gap Junction) and Joanna Lignot (Munro Language Services) for proofreading.

## Additional information

### Funding

| Funder | Grant reference number | Author |
|---|---|---|
| Fondation pour la Recherche Médicale | Graduate student Fellowship | Antoine M Valera |
| Fondation pour la Recherche Médicale | FRM 2014 DEQ20140329514 | Philippe Isope |
| Agence Nationale de la Recherche | 2010-JCJC-1403-1 MicroCer | Philippe Isope |

| European funds for Regional Development | FEDER, #A31 | Philippe Isope |
| Agence Nationale de la Recherche | ANR-13-SAMA-0010-01 CERBAIS | Philippe Isope |

The funders had no role in study design, data collection and interpretation, or the decision to submit the work for publication.

## Author contributions

AMV, Conception and design, Acquisition of data, Analysis and interpretation of data, Drafting or revising the article; FB, Conception of viral injections, Acquisition of data, Analysis and interpretation of data; SAP, Conception of viral injections, Acquisition of data; J-LD, Conception of immunohisto-chemical experiments, Contributed unpublished essential data or reagents; J-FC, Conception of viral construction and molecular biological tools, Contributed unpublished essential data or reagents; JDR, Provided essentials tools and revision of the article, Contributed unpublished essential data or reagents; BP, Analysis and interpretation of data, Contributed unpublished essential data or reagents; PI, Conception and design, Analysis and interpretation of data, Drafting or revising the article

## Author ORCIDs

Philippe Isope, http://orcid.org/0000-0002-0630-5935

## Ethics

Animal experimentation: All experimental procedures conformed to French ministry and local ethical committee (CREMEAS) guidelines on animal experimentation. All of the animals were handled according to approved institutional animal care and use committee protocols of the french ministry (#A67-2018-38)

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
