## [Decision Letter]

[Editors’ note: this article was originally rejected after discussions between the reviewers, but the authors were invited to resubmit after an appeal against the decision.]

Thank you for choosing to send your work entitled "Spatial patterns of functional synaptic connectivity in the cerebellar cortex" for consideration at *eLife*. Your full submission has been evaluated by Gary Westbrook (Senior editor), a member of our Board of Reviewing Editors, and two peer reviewers, and the decision was reached after discussions between the reviewers. Based on our discussions and the individual reviews below, we regret to inform you that your work will not be considered further for publication in *eLife*.

We recognized the value to cerebellar specialists of your demonstration of patterned input as well as measurements of the strength of connectivity. This work represents a substantial technical achievement. Nevertheless, the work was seen descriptive at this stage, lacking a body of conclusions of interest to non-specialists. Some ambiguities of interpretation and apparent experimental discrepancies were also pointed out (detailed below). The question was also raised whether blockade of GABA receptors confounded interpretations. the presentation of the manuscript and figures were hard to follow, especially for those outside the cerebellar field (for specifics, see below). Although we think this study is more suited for a more specialized journal, we hope that the comments will be useful to you.

*Reviewer #1:*

This work examines the pattern of connectivity of groups of granule cells to their targets, Purkinje (PC), Golgi, or stellate cells, in the cerebellar cortex. The approach is an interesting one: pairs of PCs are recorded with one being a reference cell and a second positioned variable distances from the reference cell. Groups of granule cells were activated by a glutamate uncaging spot that was scanned across the tissue slice in a grid pattern. The location of the midline and of zebrin labeling allowed the investigators to establish reproducible reference points that enabled pooling of data across animals. The primary conclusions are that there is indeed a pattern of projections from different granule cell areas to PCs, often very local but also across sizeable distances, that these patterns of innervation are conserved across animals but modified by activity, and that they may differ for PCs vs interneurons.

The study is impressive in its recruitment of different methods to interrogate local mapping within the cerebellum. My primary concern is that the conclusions are ultimately descriptive, i.e., there is no obvious (to me) insight about cerebellar function revealed by the establishment of these patterns of input. Given too that the work is hard to understand makes me think it may not be of general interest to an *eLife* reader.

*Reviewer #2:*

The manuscript "Spatial patterns of functional synaptic connectivity in the cerebellar cortex" by Valera et al. reports a novel and interesting finding on the strength of connections between populations of granule cells and Purkinje neurons. They map connectivity of GCs to PCs using photo-uncaging of glutamate in discreet regions of the granule cell layer and find stereotyped but heterogeneous connectivity patterns. Contrary to previous findings, granule cell populations immediately subjacent to recorded Purkinje neurons were not necessarily stronger; indeed in some cases distal GC populations showed preferred connectivity. These maps, when viewed with respect to anatomical boundaries, appear to follow loose rules and show richer connectivity patterns that other cell types such as Golgi cells and molecular layer interneurons. Together the results suggest a previously unrecognized organization of connectivity between granule cells and Purkinje neurons and a point to a role for parallel fibers in cerebellar functional circuitry where before they have appeared functionally elusive.

I find the take home message to be compelling and interesting. I do feel a deep sense of reservation that the data were recorded in the presence of GABAA receptor blockers (among many others) that could severely alter the functional relevance of the findings. The manuscript could be strengthened to better articulate the findings, justify certain analytical approaches, clarify figures, and more thoroughly address the discrepancies with previous findings. These ideas are fleshed out below:

1) Previous studies have identified a role for molecular layer interneurons in inhibiting distant Purkinje neurons through a lateral inhibition mechanism (Dizon & Khodakhah, 2011). It is therefore surprising that relatively little attention is paid to the role of inhibition in shaping GC-PC maps in the present study. Do the authors expect that the interesting heterogeneous patterns they uncovered here are robust to inhibition? Might inhibition further sculpt the patterns? How might the plane of section influence the findings (Rieubland et al. 2014)? While I think the results as presented are interesting on their own, without data on the maps with inhibition intact, it is difficult to surmise how relevant these maps are to normal physiological function of the cerebellum. At a very minimum these caveats should be more thoroughly discussed.

2) The authors should be commended in taking the initial interesting observation of structured spatial GC-PC organization to the next logical question regarding the stability of these maps after plasticity events. Yet this part of the manuscript would benefit from the most attention. I found the display of data to be a bit puzzling since the color scheme indicating changes in connectivity shifts completely from pre and post-induction. While I don't want to micromanage, it would seem that at a minimum, unchanged sites should not change color before and after induction as they currently do (black to dark gray; white to light gray, etc.). In short, the representation of data in Figure 6 should be reworked.

Similarly, the idea that motivated these experiments (that complex spatial maps of connectivity may be altered with use) is not well explored with this dataset. The authors emphasize that silent sites strengthen and strong sites weaken, but bringing the analysis back to the original point of maps is lacking. I would like to see a few lines of binaried map analysis parallel to those shown in Figure 4 for the 8 cells that underwent plasticity induction. Illustrating the before-and-after heat maps would seem to me to be essential for evaluating whether the "map" is plastic, rather than a few sites that increase or decrease their strength. Short of this, the finding that the maps are under plastic control seems weak.

3) One of the major findings is that GC-PC connectivity patterns show complex spatial patterns. On its face, this finding of course conflicts with interpretations of experiments using similar techniques that emphasize a primary role for subjacent granule cells in driving overlying Purkinje neurons. Yet the spatial map of GC connection probability in Figure 4 (and Figure 8—figure supplement 1) reveal that local GCs, in general, provide the strongest influence to overlying PCs. At first I scratched my head at this seemingly contradictory finding because the data shown in Figure 2 and Figure 6 clearly do not show strong local connectivity. Looking more closely at the Figure 4 heat map, however, it is apparent that Purkinje neurons recorded in the P1^–^ zone were unique in lacking strong local GC input. Since these cells formed the basis of several figures as examples but are different from the overall trend in their weak local input, I think the authors should highlight this difference as a finding that is otherwise obscured by the predominant effect of local GC input on the maps and in the previously published literature.

4) Relatedly, the finding that correlations in map patterns drop off with distance, (Figure 3) seems like a preliminary analysis given the pattern separation described in Figure 4. Rather than emphasize that cells nearest the midline are similar and that this similarity drops off, a pairwise cross correlation marching across the population of PCs organized spatially may offer a quantitative description of the drifting GC input pattern and an objective criterion for classifying pattern types in Figure 4. Indeed, it would be important to provide some quantitative criterion for these pattern classes. Short of that they are descriptive but not rigorously defined.

5) Summary illustrations in Figure 8 are largely ineffective and should be re-worked. The symbols atop panel A for example are puzzling: the hashtags in A are not described until the end of the B panel legend text; numbers and descriptors might be more intuitive.

[Editors’ note: what now follows is the decision letter after the authors submitted for further consideration.]

Thank you for resubmitting your work entitled "Stereotyped spatial patterns of functional synaptic connectivity in the cerebellar cortex" for further consideration at *eLife*. Your revised article has been favorably evaluated by Gary Westbrook (Senior editor), a Reviewing editor, and two reviewers. The manuscript has been greatly improved, but there are some remaining issues that need to be addressed before acceptance, as outlined below:

Both reviewers (one from the first round and one new reviewer) found the revised manuscript interesting and important, as well as improved over the first version, as indicated in their general comments, included below.

Reviewer #2:

In their revised manuscript, Valera et al. have improved upon their original manuscript by simplifying displays and performing extra analyses that bolster their conclusions. I find their report to be convincing and important. They find that the synaptic strength of parallel fibers onto Purkinje neurons, as read out by postsynaptic currents, varies along the length of the fiber. Rather than systematic changes in strength, they find highly heterogeneous synapses that nevertheless produce stereotyped patterns of synaptic strength. These patterns do not correspond to known anatomical boundaries set up by zebrin stripes. At least 4 pattern groups exist and they cluster depending upon anatomical location. The patterns reflect experience because a plasticity induction protocol changes the synaptic strength pattern. Finally, these functional synaptic strength patterns differ for Golgi cells and MLIs. This study provides a glimpse into a hitherto unknown functional organizational pattern in cerebellar cortex dictated by activity-dependent mechanisms that will almost certainly play an important role in sculpting spatiotemporal patterns of Purkinje activity during behavior.

Reviewer #3:

This paper by Philippe Isope and colleagues provides a major contribution to unraveling the functional operations of cerebellar architecture. As far as this reviewer knows this is one of the first times distant cerebellar connectivity is shown at the network level. Although somewhat descriptive, its consistent nature across animals points toward important implications.

Minor comments (both reviewers) [abridged]:

1) Introduction, end of second paragraph: The build up towards this sentence suggests that the organization of the MF input contributes to, or is even essential for the concept of microzones. It should be made clear that microzones are primarily defined based on efferent fibers and CF input (see e.g. De Zeeuw et al., 2011 Nat Rev Neur), rather than mossy fibers.

2) Results: Along the same line, the authors state that "boundaries of the functional microzones might slightly differ from a pure anatomical description". At first sight, the boundaries seem to significantly differ from those of the zebrin II pattern. Please explain the chosen wording.

3) I am missing a description of the efficacy of the cuneate/spinal cord injections. In other words, which percentage of the cuneate nucleus or of the lumbar spinal cord was colored? Was there any reason to assume heterogeneity within these nuclei in relation to the staining pattern in the cerebellar cortex? How specific were the injections (were also neighboring areas affected, for instance)?

4) Figure 2:

A) What do the authors mean with "area mapped = 664 µm" (subsection “Heterogeneous GC input maps to PCs revealed distant GC inputs“): radius or area?

B) The authors disregard the variation in stimulating at different depths within the granular layer. According to Figure 2 the variation caused by using different depths is at least equal to that caused by using different lateral distances to the recorded Purkinje cell. Please comment also on the impact of depth.

C) "No GC patch had a Z-score > 3.09 at any depth of the GC layer" should read: "al all depths"?

5) Figure 4: Several clusters appear to be slightly larger in lateral direction. I understand the cluster analysis is free, and thus these results are not hand-picked, but how do the authors interpret these extensions, including the 5th cluster attached to #4?

6) With regard to the importance of the consistency, please discuss to what extent these findings, which taken exclusively from the anterior lobe, may be extrapolated to the other parts of the cerebellum, in particular the posterior lobe. It has been shown by Gerrits and Voogd that the MFs in particularly the anterior lobe are organized in zones, whereas those in the posterior lobe are not. The patterns of MF projections in the posterior lobe are dramatically different from those in the anterior lobe.

---

## [Author Response]

[Editors’ note: the author responses to the first round of peer review follow.]

We recognized the value to cerebellar specialists of your demonstration of patterned input as well as measurements of the strength of connectivity. This work represents a substantial technical achievement. Nevertheless, the work was seen descriptive at this stage, lacking a body of conclusions of interest to non-specialists. Some ambiguities of interpretation and apparent experimental discrepancies were also pointed out (detailed below). The question was also raised whether blockade of GABA receptors confounded interpretations. the presentation of the manuscript and figures were hard to follow, especially for those outside the cerebellar field (for specifics, see below). Although we think this study is more suited for a more specialized journal, we hope that the comments will be useful to you.

Reviewer #1:

*This work examines the pattern of connectivity of groups of granule cells to their targets, Purkinje (PC), Golgi, or stellate cells, in the cerebellar cortex. The approach is an interesting one: pairs of PCs are recorded with one being a reference cell and a second positioned variable distances from the reference cell. Groups of granule cells were activated by a glutamate uncaging spot that was scanned across the tissue slice in a grid pattern. The location of the midline and of zebrin labeling allowed the investigators to establish reproducible reference points that enabled pooling of data across animals. The primary conclusions are that there is indeed a pattern of projections from different granule cell areas to PCs, often very local but also across sizeable distances, that these patterns of innervation are conserved across animals but modified by activity, and that they may differ for PCs vs interneurons.*

*The study is impressive in its recruitment of different methods to interrogate local mapping within the cerebellum. My primary concern is that the conclusions are ultimately descriptive, i.e., there is no obvious (to me) insight about cerebellar function revealed by the establishment of these patterns of input. Given too that the work is hard to understand makes me think it may not be of general interest to an eLife reader.*

The mesoscopic level of organization, i.e. the communication between microcircuits such as modules (microzones in our case) illustrates how a brain area coordinates computation made by individual modules. A critical point is to understand how information is processed in individual modules and communicated to other modules. Parallel or distributed processing are certainly underpinned by different functional organization of neuronal networks. In the barrel cortex of rodents, initial suggestion that information from individual whiskers are processed mostly in one cortical column and combined at a later stage was largely broken by data demonstrating that intermodular communication can drive modular processing (see for example Estebanez et al., 2012 Nat Neurosci. 2012 Dec;15(12):1691-9.). Furthermore, as written now in the Introduction of the manuscript, "whether neighboring modules strictly follow the same functional and cellular organization rules or display subtle differences in local architecture or neurochemical expression that would determine module-specific function or processing rules is not established".

In the cerebellar cortex, if anatomical modules are well defined, subtle differences in the functional anatomy and the physiology have been shown that lead to the idea that each module might have its own identity. Our results demonstrate that these individual modules communicate together and collaborate in order to process information. Moreover this functional link can be tuned but is conserved between animals demonstrating that it is a rule not a simple selection of synaptic inputs. We think that the observed patterns actually highlight new operational rules for the coordination of information in the cerebellar cortex that might illustrate a more general mechanism. A similar description has been performed for example in Cossell L et al. (Nature. 2015 Feb 19;518(7539):399-403.). Although our observations are made in the cerebellar cortex, cell-specific connectivity profiles underlying inter-modular and intra modular information processing might be of general interest, since modular systems are present in many cortical structures. Therefore, we think that instead of being descriptive, our work provides the proof of concept for the activity dependent (as shown in the new Figure 6) selection of conserved sets of synapses that underlie modular communication. More specifically, our findings demonstrate that:

1) Granule cells (the input stage of the cerebellar cortex) originating in different functional microzones (modules) communicate their information through the parallel fiber excitatory pathway to Purkinje cells from local and distant microzones in a specific and reproducible manner. Granule cell input maps are specific for one type of connection since granule cell-Purkinje cell and granule cell-molecular interneuron synapses define different maps. Also, the fact that one type of synapse, the granule cell-Golgi cell synapse, does not participate in the communication between microzones, is a surprising result in regard to the morphology of this interneuron, and an important new characteristic of the cerebellar cortical network.

2) These granule cell input maps are conserved from animal to animal indicating that one given area of the granule cell layer dealing with specific mossy fiber inputs contacts preferentially identified microzones, linking the microzones through the parallel fiber pathway. This illustrates how local association rules between cerebellar modules could influence the coordinated output of the cerebellar cortex. On a more practical point of view, the fact that these patterns are reproducible, and that their position is easy to identify (the midline is easy to find even without zebrin markers) and relatively easy to observe (lobule IV is accessible to in vivo imaging), give them a great potential for any future studies interested in studying how modules coordination works in vivo and in vitro.

This is one of the first demonstration that functional synaptic connectivity can be specifically conserved between animals.

3) These results definitely answer one of the longest controversy in the field: the organization of synaptic functional organization of granule cell to Purkinje cell connection. By monitoring Purkinje cell discharge in vivo, several groups suggested that only local granule cells (through the ascending part of the granule cell axon) can drive Purkinje cells firing (see Bower and Woolston, J Neurophysiol. 1983 Mar;49(3):745-66; Cohen and Yarom, Proc Natl Acad Sci U S A. 1998 Dec 8;95(25):15032-6.) indicating that information in the cerebellar cortex might be process in parallel and that coordination between microzones might occur at a downstream level, maybe outside the cerebellum. Conversely, other groups suggested that Purkinje cells are only driven by one restricted group of granule cells, but not local (for review Nat Rev Neurosci. 2010 Jan;11(1):30-43.). However, studies performed in vivo, monitored the discharge of the Purkinje cells following stimulation of cerebellar inputs, a very indirect assessment of granule cell-Purkinje cell functional organization. Ultimately, based on in vitro uncaging techniques, another group suggested that granule cells give a broad input onto Purkinje cells (Walter et al., 2009 J Neurosci. 2009 Jul 1;29(26):8462-73.). However, they did not consider the location of the recordings in their study, smoothing heterogeneities.

Our data provide a direct proof conciliating both hypotheses: distant microzones communicate through parallel fibers in the cerebellar cortex and local granule cells provide a strong input to Purkinje cells.

The strength of our study is that it extents classical synaptic observations to the level of the microcircuit organization, by recording targeted anatomical modules.

Reviewer #2:

*[…] I find the take home message to be compelling and interesting. I do feel a deep sense of reservation that the data were recorded in the presence of GABAA receptor blockers (among many others) that could severely alter the functional relevance of the findings. The manuscript could be strengthened to better articulate the findings, justify certain analytical approaches, clarify figures, and more thoroughly address the discrepancies with previous findings. These ideas are fleshed out below: 1) Previous studies have identified a role for molecular layer interneurons in inhibiting distant Purkinje neurons through a lateral inhibition mechanism (Dizon & Khodakhah, 2011). It is therefore surprising that relatively little attention is paid to the role of inhibition in shaping GC-PC maps in the present study. Do the authors expect that the interesting heterogeneous patterns they uncovered here are robust to inhibition? Might inhibition further sculpt the patterns? How might the plane of section influence the findings (Rieubland et al. 2014)? While I think the results as presented are interesting on their own, without data on the maps with inhibition intact, it is difficult to surmise how relevant these maps are to normal physiological function of the cerebellum. At a very minimum these caveats should be more thoroughly discussed.*

We thank the reviewer for raising this important point. We acknowledge and anticipate that the granule cell-MLI-Purkinje cell feedforward pathway and the off-beam inhibition through basket cells will result in a net effect that might differ from our excitatory maps, and we now discuss this point in the text. However, it is important to keep in mind that inhibition won't change in any way the structure of the granule input maps that we have described here, which are only excitatory connection maps. We also would like to insist on the fact that distant granule cells hardly contact local Golgi cells and that the axon of Golgi cells is restricted to the local area. Consequently, feedback inhibition is restricted to local area which will affect only local granule cells. The excitatory communication lines described here between microzones are thus physiologically relevant even if the final effect on Purkinje cell discharge is ultimately linked to the balance between direct excitation from granule cells and disynaptic inhibition from molecular layer interneurons.

As now stated in the Discussion, recordings were intentionally performed with GABAA antagonist as well as blocker of plasticity in order to determine the precise organization of the granule cell inputs to Purkinje/MLI and Golgi cells.

Nevertheless, studying the inhibitory input map would be interesting for the prediction of the physiological outcomes of information processing, however RuBi-glutamate or MNI-glutamate partially block GABAA receptors (see Fino et al., 2011), then another set of techniques should be used to tackle this question.

Again, I would like to emphasize that inhibition, although physiologically relevant, won't change the outcomes of this study, since distant microzones will be connected together through granule cell-Purkinje cell synapses or by the way granule cell-MLI-Purkinje cell synapses. Our striking results demonstrating that *functional synaptic connectivity can be conserved between animals* won't be affected by inhibition. Therefore the cortical coordination of microzones by granule cells is a robust phenomenon.

*2) The authors should be commended in taking the initial interesting observation of structured spatial GC-PC organization to the next logical question regarding the stability of these maps after plasticity events. Yet this part of the manuscript would benefit from the most attention. I found the display of data to be a bit puzzling since the color scheme indicating changes in connectivity shifts completely from pre and post-induction. While I don't want to micromanage, it would seem that at a minimum, unchanged sites should not change color before and after induction as they currently do (black to dark gray; white to light gray, etc.). In short, the representation of data in Figure 6 should be reworked.*

We agree, this figure has been rebuilt and focused on the modification of the granule cell input map after plasticity induction (see next point) as rightly pointed out by the reviewer.

*Similarly, the idea that motivated these experiments (that complex spatial maps of connectivity may be altered with use) is not well explored with this dataset. The authors emphasize that silent sites strengthen and strong sites weaken, but bringing the analysis back to the original point of maps is lacking. I would like to see a few lines of binaried map analysis parallel to those shown in Figure 4 for the 8 cells that underwent plasticity induction. Illustrating the before-and-after heat maps would seem to me to be essential for evaluating whether the "map" is plastic, rather than a few sites that increase or decrease their strength. Short of this, the finding that the maps are under plastic control seems weak.*

The group of cells experiencing plasticity did not allow us to rebuild a map for all 4 clusters, but 7 cells recorded in Cluster 1 (see text and Figure 4) accurately reproduced the GC input pattern. Therefore, we reanalyzed the dataset and described the median histogram of the z-score of charge (Figure 6) for this group of Purkinje cells. We then demonstrated that a new group of connected sites appeared at a specific position in the mediolateral axis (Figure 6). We then concluded that the map has changed, suggesting that plastic modification can tune the map. As expected from a stimulation protocol targeting granule cells at random location, the effect was particularly obvious in regions where granule cells were initially silent.

*3) One of the major findings is that GC-PC connectivity patterns show complex spatial patterns. On its face, this finding of course conflicts with interpretations of experiments using similar techniques that emphasize a primary role for subjacent granule cells in driving overlying Purkinje neurons. Yet the spatial map of GC connection probability in Figure 4 (and Figure 8—figure supplement 1) reveal that local GCs, in general, provide the strongest influence to overlying PCs. At first I scratched my head at this seemingly contradictory finding because the data shown in Figure 2 and Figure 6 clearly do not show strong local connectivity. Looking more closely at the Figure 4 heat map, however, it is apparent that Purkinje neurons recorded in the P1^–^ zone were unique in lacking strong local GC input. Since these cells formed the basis of several figures as examples but are different from the overall trend in their weak local input, I think the authors should highlight this difference as a finding that is otherwise obscured by the predominant effect of local GC input on the maps and in the previously published literature.*

This is perfectly right, we have adjusted consequently. As mentioned in the main text, distant hotspots of connected granule cells are sometimes stronger than the local input. Also, as pointed out by the reviewer, one group of cells show no (or weak) local inputs (however, it represents less than 10% of the cells). This point has been clarified.

*4) Relatedly, the finding that correlations in map patterns drop off with distance, (Figure 3) seems like a preliminary analysis given the pattern separation described in Figure 4. Rather than emphasize that cells nearest the midline are similar and that this similarity drops off, a pairwise cross correlation marching across the population of PCs organized spatially may offer a quantitative description of the drifting GC input pattern and an objective criterion for classifying pattern types in Figure 4. Indeed, it would be important to provide some quantitative criterion for these pattern classes. Short of that they are descriptive but not rigorously defined.*

We agreed with this point and decided to remove original Figure 3 and Figure 4. The binarization has been abandoned because a simpler and more straightforward analysis lead to similar results. As suggested by the reviewer, a correlation matrix has been built (see Figure 4) using groups of Purkinje cells in order to minimize the noise and extract the median pattern at a given position of the granular layer. Regions that displayed similar GC input patterns in our previous analysis (old Figure 4) are now highly correlated in the correlation matrix. This approach is more precise and quantitative. We also applied a bi-clustering algorithm (see Methods) that allowed us to extract the boundaries of clusters of neighboring Purkinje cells. We hope that these two analyses highlight the main result of the manuscript: the stereotyped organization of granule cell inputs.

*5) Summary illustrations in Figure 8 are largely ineffective and should be re-worked. The symbols atop panel A for example are puzzling: the hashtags in A are not described until the end of the B panel legend text; numbers and descriptors might be more intuitive.*

This figure was clearly not optimal. We have completely redone it.

[Editors’ note: the author responses to the re-review follow.]

Minor comments (both reviewers) [abridged]:

*1) Introduction, end of second paragraph: The build up towards this sentence suggests that the organization of the MF input contributes to, or is even essential for the concept of microzones. It should be made clear that microzones are primarily defined based on efferent fibers and CF input (see e.g. De Zeeuw et al., 2011 Nat Rev Neur), rather than mossy fibers.*

We agree and have revised the manuscript accordingly (Introduction, second paragraph).

*2) Results: Along the same line, the authors state that "boundaries of the functional microzones might slightly differ from a pure anatomical description". At first sight, the boundaries seem to significantly differ from those of the zebrin II pattern. Please explain the chosen wording.*

The wording was inappropriate. The sentence has now been rewritten: "...boundaries of the functional organization between microzones differ from a pure anatomical description."

*3) I am missing a description of the efficacy of the cuneate/spinal cord injections. In other words, which percentage of the cuneate nucleus or of the lumbar spinal cord was colored? Was there any reason to assume heterogeneity within these nuclei in relation to the staining pattern in the cerebellar cortex? How specific were the injections (were also neighboring areas affected, for instance)?*

We thank the reviewers for pointing this out. We have now included a description of the injection sites in the Methods section and corrected a mistake in the description of the spinal cord injection. A supplementary figure has also been added.

In brief, our injections were mostly focused on cuneate injections (4 animals have been injected with AAV2/1-GFP and 2 with fluoro-ruby). In 4 out of 6 animals, injection site spread into the entire lateral and dorso-ventral axes and 350-400 µm in the antero-posterior axis (see Figure 1—figure supplement 2) of the external cuneate. Cortical projections in lobule III and IV have been measured in 13 different slices in these 4 animals. Also, one large injection (1.2 µl) injection in the lumbar segments of one animal has been performed in the spinal cord which led to an infected area ≈1 cm centered around L3-L5 (see Figure 1—figure supplement 2). Cortical projections have been measured in 4 different cerebellar slices.

Although we cannot guarantee that the entire cuneate nucleus has been injected, we note that the size and the position of the bands in the cerebellar cortex are very similar between the 4 different animals (cuneate injections) and that they match perfectly with all the previous studies. Please see Figure 8D of Gebre SA, Reeber SL, Sillitoe RV (Brain Struct Funct. 2012 Apr;217(2):165-80) or Figure 9 of Ji and Hawkes (Neuroscience. 1994 Aug;61(4):935-54): the cuneate and spinocerebellar projections in lobule III/IV can be perfectly superimposed to our Figure 1. See also Figure 1 of Huang CC et al., Elife. 2013 Feb 26;2:e00400 for cuneate projections.

We therefore postulate that the boundaries of the mossy fiber projections observed in our experiments represent anatomical limit of specific group of mossy fiber inputs originating in the cuneate nucleus of the spinal cord.

4) Figure 2:

*A) What do the authors mean with "area mapped = 664 µm" (subsection “Heterogeneous GC input maps to PCs revealed distant GC inputs“): radius or area?*

This is the width (in the mediolateral axis) of the granule cell layer mapped using our photostimulation system. The term "area" has been corrected for width.

*B) The authors disregard the variation in stimulating at different depths within the granular layer. According to Figure 2 the variation caused by using different depths is at least equal to that caused by using different lateral distances to the recorded Purkinje cell. Please comment also on the impact of depth.*

We agree that GC inputs varied at different depth of the GC layer as well as in the medioloateral axis (although a distal response could often be observed at several depth of the granular layer). However, since our study focused on the communication between distant microzones, we considered that one connected site (a third of the depth of the GC layer in Figure 2) to a given microzone is sufficient to consider that some information can be communicated between the 2 microzones. For this reason, we choose to attribute the maximum GC inputs observed in the entire depth of a GC column to the mediolateral position. Furthermore, mossy fiber from one specific nucleus project to all depths of the GC layer, at identified mediolateral position (see our paper and aforementioned articles in point 3). Therefore, two adjacent GC sites in the mediolateral axis may carry information from different type of mossy fiber inputs while sites at different depth of GC layer are likely to convey information from similar sources. This point is mentioned in the text (subsection “Heterogeneous GC input maps to PCs revealed distant GC inputs”).

*C) "No GC patch had a Z-score > 3.09 at any depth of the GC layer" should read: "al all depths"?*

Yes, this has been corrected.

5) Figure 4: Several clusters appear to be slightly larger in lateral direction. I understand the cluster analysis is free, and thus these results are not hand-picked, but how do the authors interpret these extensions, including the 5th cluster attached to #4?

Indeed, the biclustering algorithm has been applied without assumption of shape or size for the clusters. In the matrix, the median pattern of groups of PCs in a 100 µm window has been correlated. Since two adjacent groups are separated by only 10 µm, they share some patterns. Therefore, several lines of the matrix are necessary to separate different profiles. For this reason, clusters based only on the correlation matrix are larger, explaining the extensions from the result of the biclustering algorithm. Regarding the 5th cluster, because it is situated at the edge of the field (>300 µm from the midline), it contains fewer patterns and has not been detected by the biclustering algorithm.

*6) With regard to the importance of the consistency, please discuss to what extent these findings, which taken exclusively from the anterior lobe, may be extrapolated to the other parts of the cerebellum, in particular the posterior lobe. It has been shown by Gerrits and Voogd that the MFs in particularly the anterior lobe are organized in zones, whereas those in the posterior lobe are not. The patterns of MF projections in the posterior lobe are dramatically different from those in the anterior lobe.*

We agree that the overall topographical organization of the posterior lobe is different than the anterior lobe with zebrin positive bands wider and zebrin negative bands much thinner. However, anatomical evidences suggest that a microzonal organization can also be observed in the posterior lobe. This has been reviewed for example by Jan Voogd several times (Voogd, J. (2014). Front. Syst. Neurosci. *8*, 1–14.). The group of Richard Apps also described the organization of discrete modules in the posterior lobe (Cerminara, N.L., Aoki, H., Loft, M., Sugihara, I., and Apps, R. (2013). J. Neurosci. *33*, 16427–16442.). Matsuo Matsushita showed that mossy fibers can project in bands in lobule VIII (Figures 14-15 of Yaginuma and Matsushita J Comp Neurol. 1987 Apr 1;258(1):1-27). Interestingly mossy fibers projecting to anterior lobule III/IV make collaterals to posterior lobule VIII suggesting that common mechanisms may apply to this lobule (Voogd, J., Pardoe, J., Ruigrok, T.J.H., and Apps, R. (2003). J. Neurosci. *23*, 4645–4656). Although anatomical differences are obvious, we suggest that communication between microzones through PFs could be a general mechanism involved in information processing in the cerebellum.

However, whether conserved and stereotyped patterns across animals will be the rule in other part of the cerebellum needs to be demonstrated. For example, in crus I/II, it is possible that animal specific processing linked to vibrissa tactile discrimination leads to unique communication rules, as sensory discrimination relies on individual history.

This statement has been added to the Discussion (subsection “The functional units process information from different origins”, second paragraph).